# Knowledge-enhanced visual-language pre-training on chest radiology images

**Xiaoman Zhang** [1,2], **Chaoyi Wu**[1,2], **Ya Zhang**[1,2], **Weidi Xie** [1,2] ✉ & **Yanfeng Wang** [1,2] ✉

While multi-modal foundation models pre-trained on large-scale data have been successful in natural language understanding and vision recognition, their use in medical domains is still limited due to the fine-grained nature of medical tasks and the high demand for domain knowledge. To address this challenge, we propose an approach called Knowledge-enhanced Auto Diagnosis (KAD) which leverages existing medical domain knowledge to guide vision-language pre-training using paired chest X-rays and radiology reports. We evaluate KAD on four external X-ray datasets and demonstrate that its zero-shot performance is not only comparable to that of fully supervised models but also superior to the average of three expert radiologists for three (out of five) pathologies with statistical significance. Moreover, when few-shot annotation is available, KAD outperforms all existing approaches in fine-tuning settings, demonstrating its potential for application in different clinical scenarios.

Foundation models[1], such as BERT[2], GPT[3] and CLIP[4], have shown great promise in feature transfer and generalization to a wide spectrum of downstream tasks[5–8], for example, in natural language processing or computer vision. However, the development of foundation models in medical domains has largely lagged behind[4,9–11]. A direct extension of existing approaches[12,13] that align visual and text modalities in the medical domain hardly generalizes toward diseases or radiology findings beyond those seen at training. This is largely due to the requirement for fine-grained recognition in medical tasks (i.e., the clues for medical diagnosis often lie in subtle and regional signals), as well as the abstractness of many complex and professional medical terminologies (e.g., infiltrates refers to the white spots in the lungs). Therefore, to effectively model the intricate and specialized concepts of medical applications, domain knowledge is indispensable.

In this paper, we aim to build a foundation model for chest X-rays by training on paired images and reports[14], termed Knowledge-enhanced Auto Diagnosis (KAD). As illustrated in Fig. 1, unlike existing approaches that simply align the image to raw textual reports, we investigate various ways to extract information from the given reports and explicitly leverage a well-established medical knowledge graph to train the knowledge encoder. In specific, the proposed KAD follows a two-stage framework; first, we learn a neural representation of

knowledge graph, with the entities represented as nodes, and relations between them as edges, that offers scaffolds for structured, multi-step reasoning about entities; second, we extract the clinical entities and relations from radiology reports in three different ways, for example, by heuristically defined rules, by using RadGraph, or by using ChatGPT, then, we exploit the pre-trained knowledge encoder to guide the visual representation learning using image and radiology reports, effectively injecting the domain knowledge into the visual encoder. Architecturally, to enable flexible zero-shot evaluation on arbitrary diseases or radiology findings, we utilize a query-based transformer architecture, termed as Disease Query Network (DQN), that can take the disease name as 'query', iteratively attending the visual feature to get the model prediction, and the attention map provides reliable visual evidence for clinical decision.

Here, to demonstrate the effectiveness of the proposed KAD, we experiment on four external X-ray datasets, i.e., PadChest[15], ChestXray14[16], CheXpert[17] and ChestX-Det10[18]. KAD is shown to be superior in auto-diagnosis for pathologies that are unseen in the training procedure, with zero-shot performance significantly higher than existing state-of-the-art medical visual-language models for 193 pathologies on PadChest, and even comparable to the fully supervised approaches. To the best of our knowledge, this is the first model pre-

[1]Cooperative Medianet Innovation Center, Shanghai Jiao Tong University, 200240 Shanghai, China. [2]Shanghai Artificial Intelligence Laboratory, 200232 Shanghai, China. ✉e-mail: weidi@sjtu.edu.cn; wangyanfeng@sjtu.edu.cn

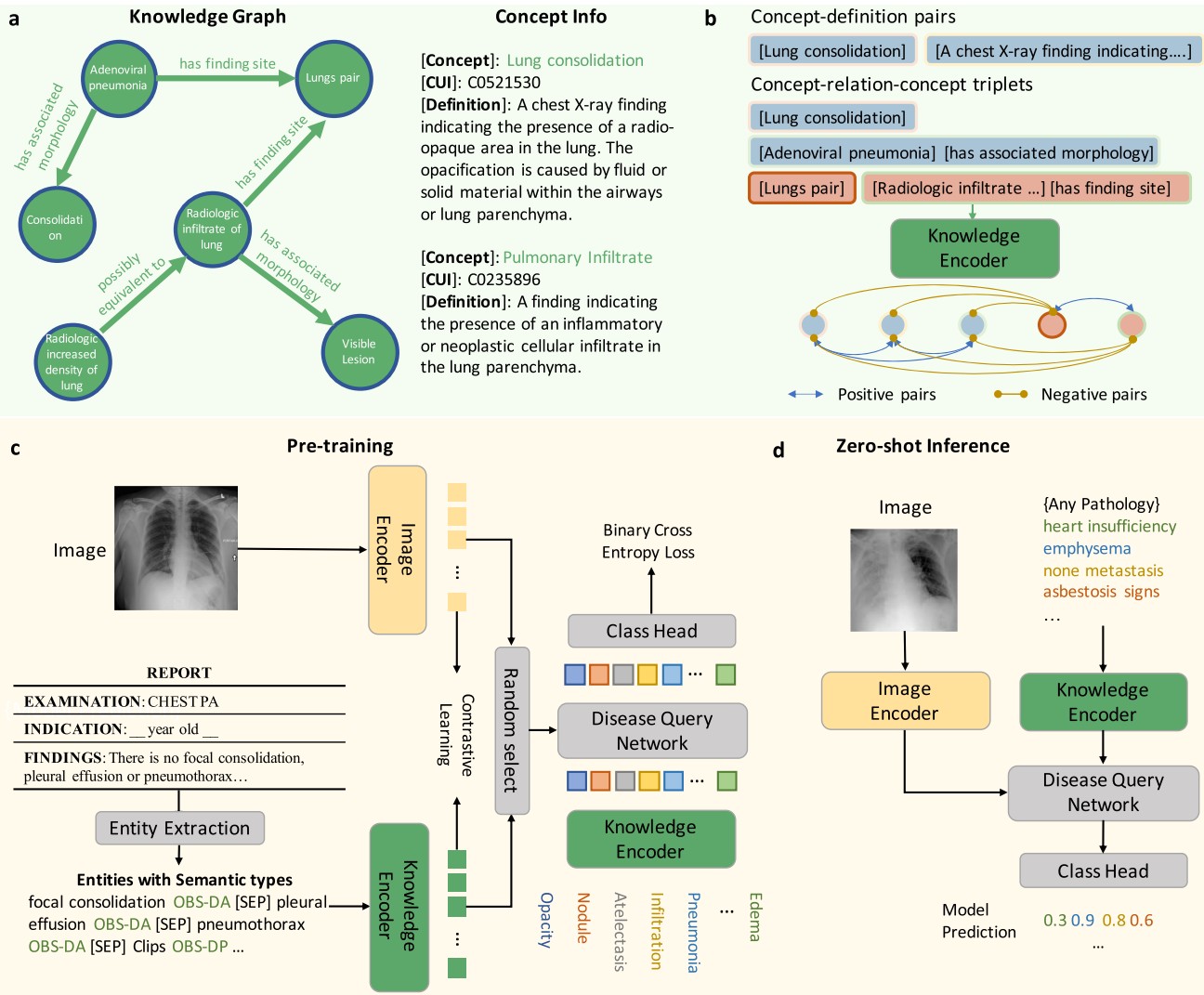

**Fig. 1 | Overview of the KAD workflow. a** Knowledge base used for training the knowledge encoder. It contains two parts, a knowledge graph consisting of concept-relation-concept triplets and a concept info list consisting of concept-definition pairs. **b** The knowledge encoder is trained to learn textual representations by maximizing similarities between positive pairs. **c** We first extract the clinical entities and relations from the radiology reports; this can be achieved by heuristic rules, using an off-the-shelf reports information extraction toolbox (Entity Extraction), or ChatGPT, then we employ the pre-trained knowledge encoder to perform image-text contrastive learning with paired chest X-rays and extracted entities and optimize a Disease Query Network (DQN) for classification. **d** During the inference stage, we simply encode the disease name as a query input, and DQN will output the probability that the pathology is present in the input image.

trained with an X-ray image and report and demonstrates comparable or superior performance to the average for expert radiologists on CheXpert. In addition, when additional manual annotations are available, our model can also be fine-tuned in the same protocol as existing self-supervised learning methods, with further boosted performance, demonstrating its superior transferability. We believe that this work presents a promising idea for domain knowledge injection in developing foundation models for AI-assisted diagnosis in radiography.

## Results

### Overview

The goal of our proposed Knowledge-enhanced Auto Diagnosis (KAD) is to improve vision-language pre-training and facilitate the auto-diagnosis for chest X-ray images by leveraging domain knowledge, for example, in existing knowledge graph (Unified Medical Language System, UMLS)[19], pre-defined heuristic rules, or off-the-shelf reports information extraction toolbox (RadGraph)[20]. In this section, all baselines are pre-trained on the MIMIC-CXR[14] dataset and directly evaluated on four well-established multi-center datasets, i.e., zero-shot inference, including PadChest[15], NIH ChestX-ray[16], CheXpert[17], and ChestX-Det10[18]. In all cases, the model makes inferences by predicting whether the queried disease exists in the input image.

Note that at the core of our study is to develop a pre-training procedure for incorporating medical domain knowledge, for example, UMLS, RadGraph, etc.; such design naturally equips KAD with additional information than existing self-supervised approaches. However, from a practical perspective, KAD training only exploits off-the-shelf tools and thus is equally scalable to large datasets as self-supervised learning approaches do, while demonstrating significantly superior performance on identifying diseases not encountered at training time and handling long-tail recognition problems. We conduct extensive ablation studies to analyze the contribution of certain model components and the impact of image resolution and visual backbone.

### PadChest

We evaluate KAD on the diagnosis task in the PadChest dataset, where the images are labeled with 174 different radiographic findings and 19 differential diagnoses. The fundamental challenge of the PadChest

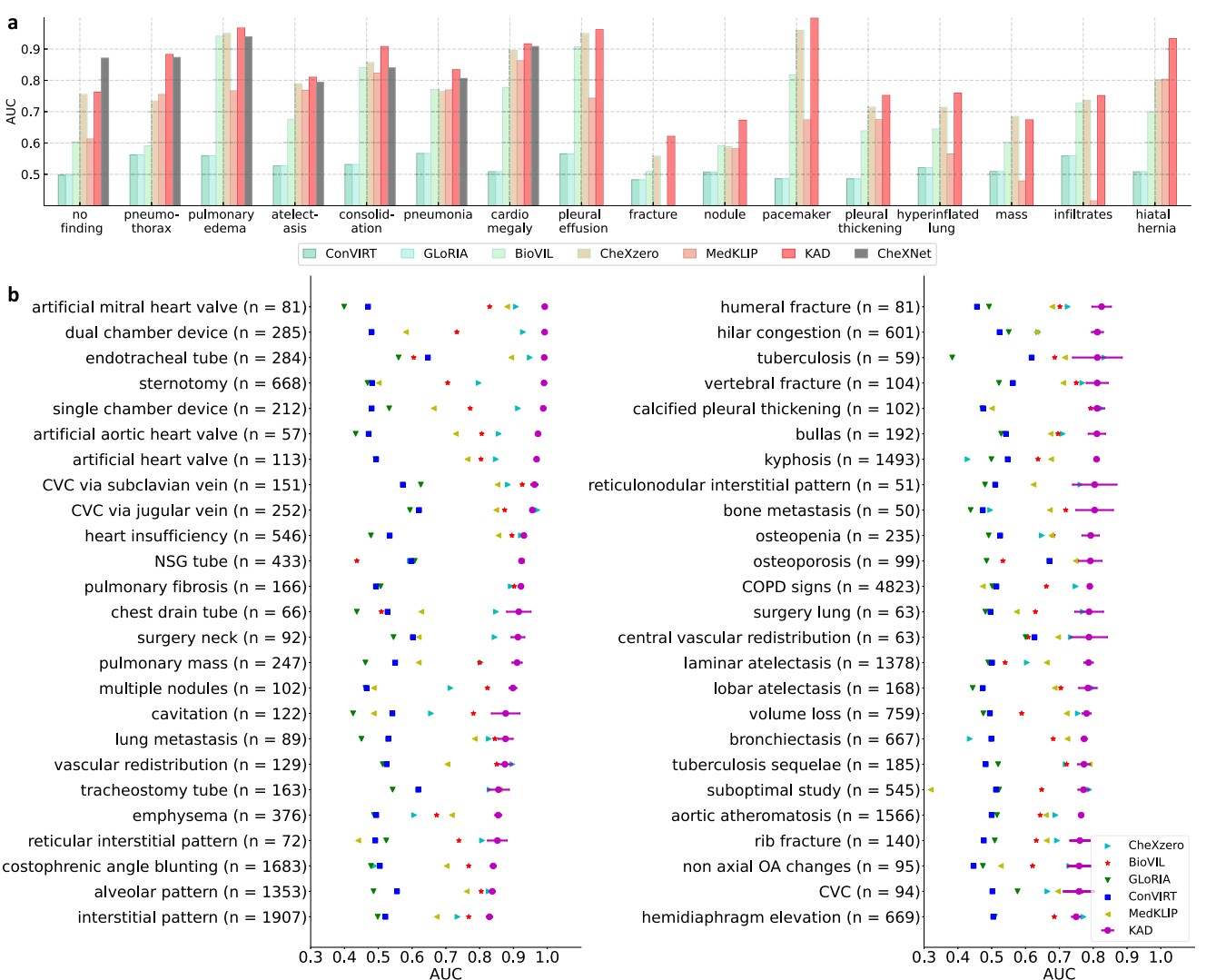

**Fig. 2 | Comparison of KAD with SOTA medical image-text pre-training models under zero-shot setting on radiographic findings or diagnoses in the PadChest dataset.** We evaluate model on the human-annotated subset of the PadChest dataset ($n = 39{,}053$ chest X-rays), and mean AUC and 95% CI of KAD are shown for each radiographic finding or diagnosis ($n > 50$). **a** Results of seen classes. Note that CheXNet is a supervised model trained on the PadChest dataset. **b** Results of unseen classes. KAD achieves an AUC of at least 0.900 on 31 classes and at least 0.700 on 111 classes out of 177 unseen classes in the PadChest test dataset. Top 50 classes where ($n > 50$) in the test dataset ($n = 39{,}053$) are shown in the figure.

dataset lies in the long-tailed class distribution, as illustrated in Supplementary Fig. 1, where only 21 classes have more than 1000 samples, and only 16 classes are seen during pre-training. Here, we denote the diseases that are seen by Disease Query Network (DQN) at the model training stage as seen diseases and the others as unseen diseases. As shown in Fig. 2b, KAD outperforms existing SOTA models on most unseen radiographic findings, and the model achieves an AUC of at least 0.900 on 31 classes and at least 0.700 on 111 classes out of 177 unseen classes in the PadChest test dataset ($n = 39{,}053$) Supplementary Figs. 2–5 include results on all 193 radiographic findings and diagnoses. As shown in Fig. 2a, while comparing the performance with fully supervised model, we observe that KAD significantly outperforms CheXNet[21] on five out of the seven pathologies diagnosis, with AUC of 0.809 (95% confidence interval (CI) 0.796, 0.822) for atelectasis, 0.916 (95% CI 0.913, 0.919) for cardiomegaly, 0.910 (95% CI 0.896, 0.924) for consolidation, 0.966 (95% CI 0.958, 0.974) for edema and 0.835 (95% CI 0.829, 0.842) for pneumonia; additionally, KAD demonstrates superior performance than other existing self-supervised visual-language models using image-text pairs, for example, AUC 11% higher on pneumothorax than CheXzero[22].

## ChestXray14

Figure 3 and Supplementary Tables 1 and 2 present results of KAD and other approaches on the NIH ChestXray14 dataset, where the images are labeled with 14 different diseases, including only one unseen class "fibrosis", not appearing in any of the reports used for training. We conduct extensive experiments on both zero-shot and fine-tuning settings; in the latter case, we verify the model's performance by varying the percentage of images for fine-tuning. As shown in Supplementary Table 1, KAD achieves the highest performance over all existing self-supervised visual-language models trained with image-text pairs; for instance, it gives an average AUC of 0.786 (95% CI 0.770, 0.808). For 13 of the 14 pathologies, KAD gets significantly higher AUC than the best-performing baseline. Additionally, in the fine-tuning scenario shown in Supplementary Table 2, KAD also demonstrates substantial improvements in all evaluation metrics over the existing approaches. As we can see, KAD consistently maintains large advantages over other methods under different labeled conditions and exhibits the largest performance improvements with respect to these baselines when only 1% data is used for training; for example, KAD surpasses ConVIRT by about 13.8%, and the widely adopted ImageNet-

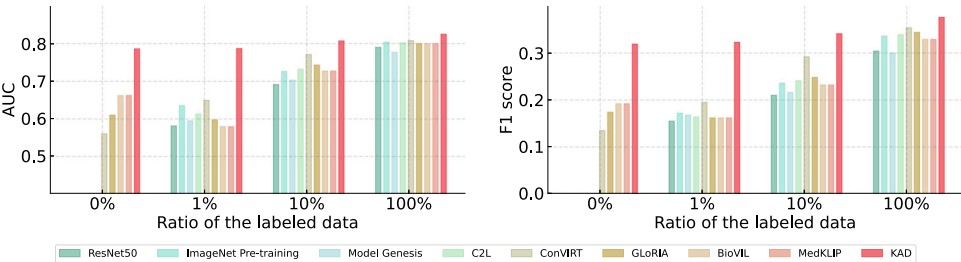

**Fig. 3 | Comparison of proposed KAD with SOTA self-supervised baseline models and medical image-text pre-training models on ChestXray14 with different ratio of labeled data used for fine-tuning.** AUC, F1 score are reported, and the metrics refer to the macro average on all the diseases. Note that for fairness, all baselines use the same backbone as the basic image encoder (that is, ResNet50). The percentages refer to the percentage of labels used in the training data.

based pre-training by about 15.2% on average, with largest improvements on hernia (>29%) and edema (>10%), especially under limited supervision.

### CheXpert

Figure 4 and Supplementary Table 3 present experimental results from KAD and other image-text pre-training models; the performance is compared to three human radiologists on the CheXpert test set. The results show that KAD consistently outperforms existing approaches on the average evaluation scores, demonstrating strong generalization ability. When compared with the mean performance of three radiologists, it is notable that KAD achieves statistically higher mean Matthews Correlation Coefficient (MCC) metric on the majority of the competition pathologies. In particular, on atelectasis (KAD 0.613 (95% CI 0.567, 0.659) vs. Radiologists 0.548), edema (KAD 0.666 (95% CI 0.608, 0.724) vs. Radiologists 0.507) and pleural effusion (KAD 0.702 (95% CI 0.653, 0.751) vs. Radiologists 0.548). On consolidation (Radiologists 0.359 (95% CI 0.262, 0.444) vs. KAD 0.357), our model is slightly worse, but not statistically significantly. On the F1 metric, there is no statistically significant difference between the mean performance from KAD 0.647 (95% CI 0.591, 0.702) and that of the radiologists 0.619 (95% CI 0.585, 0.642), with only the exception of edema, where KAD performs significantly better (KAD 0.701 (95% CI 0.647, 0.754) vs. Radiologists 0.583).

### ChestX-Det10

Figure 5 shows the results for zero-shot diagnosis and grounding on the ChestX-Det10 test set. Our proposed model outperforms existing methods on majority and mean values. Specifically, KAD obtains significant improvements over GLoRIA[9] in the difficult categories, i.e., calcification (>7%) and fracture (>18%). Additionally, it is worth noting that the ability to localize disease is significantly improved with the image resolution increasing, especially for some diseases with small bounding boxes; KAD-1024 shows not only higher accuracy but also better precision. Last but not least, we provide a thorough ablation study of KAD in Tables 1 and 2. More details can be found in the "Methods".

### Discussion

The purpose of this work was to develop a foundation model for chest X-rays by exploiting the existing knowledge prior in the medical domain. Here, we provide a discussion to analyze the performance of KAD from different perspectives.

KAD achieves comparable results with expert radiologists. While comparing KAD to human expert radiologists on CheXpert, the results in Fig. 4 indicate that the model can keep up or even surpass the performance of experienced clinicians in some diagnostic tasks, i.e., KAD achieves statistically higher MCC on the average of five pathologies, over the mean performance of three radiologists, demonstrating the effectiveness of knowledge-enhanced pre-training. As a result, the proposed KAD model can serve as a reference when disagreement occurs among physicians.

KAD achieves comparable results to supervised models. As shown in Fig. 2a, the performance of KAD is comparable to, and sometimes exceeds, fully supervised methods on PadChest; for example, KAD brings a 6.8% performance gain on "consolidation". Note that KAD achieves such results without seeing any training images from PadChest, which is collected in a different country from the training dataset MIMIC-CXR. Additionally, as shown in Fig. 2b, the application of a fully supervised model is limited to a closed set of categories, while KAD allows the query input to be arbitrary pathologies, identifying radiological findings with high accuracy on PadChest, demonstrating strong generalization ability and robustness to various clinical diagnosis categories.

KAD enables superior results on zero-shot diagnosis. By establishing connections between images and texts, vision-language models are able to turn visual classification into zero-shot inference beyond the limited set of pathologies. In contrast to existing approaches that directly match raw reports with image scans, we leverage domain knowledge from a well-structured knowledge graph (UMLS) and preprocess the reports by extracting medical entities. As in the report, the same medical concept can appear in various forms, for example, nonstandard names, abbreviations, and misspellings in the reports, which may lead to noisy supervision. While our proposed knowledge-enhanced mechanism enables the establishment of relations between different radiological concepts, thus enabling generalization to unseen classes, as suggested by the results on PadChest (Fig. 2), KAD enables the identification and diagnosis of 177 unseen classes of radiological findings with high accuracy.

KAD enables data-efficient transfer across various tasks. When manual annotations for target downstream datasets are available, KAD consistently outperforms the existing visual-language models under different fine-tuning settings, i.e., using a variable number of data for fine-tuning, as shown in Fig. 3. Note that, in the following classes, including cardiomegaly, effusion, pneumonia, pneumothorax, and hernia, KAD required only 1% labeled data to achieve better results than others under 100% label ratio, showing the potential of KAD in data-efficient transfer learning. Such a phenomenon indicates that KAD can fully inherit the benefits from pre-training, i.e., leveraging both the DQN module and the pre-trained text encoder.

KAD provides a grounded heatmap for making clinical decisions. In addition to auto-diagnosis, explainability is equally critical in AI-assisted applications, as it may help clinicians to discover and understand the evidence that the AI algorithm bases its predictions on, potentially presenting transparency and enabling better collaboration in the clinical setting. Here, we qualitatively analyze KAD by averaging the cross-attention map in each transformer layer in DQN and visualizing the results of KAD with different resolutions in Fig. 6. It can be observed that the model is indeed pooling information from regions that well match radiologists' diagnoses on different pathologies, i.e.,

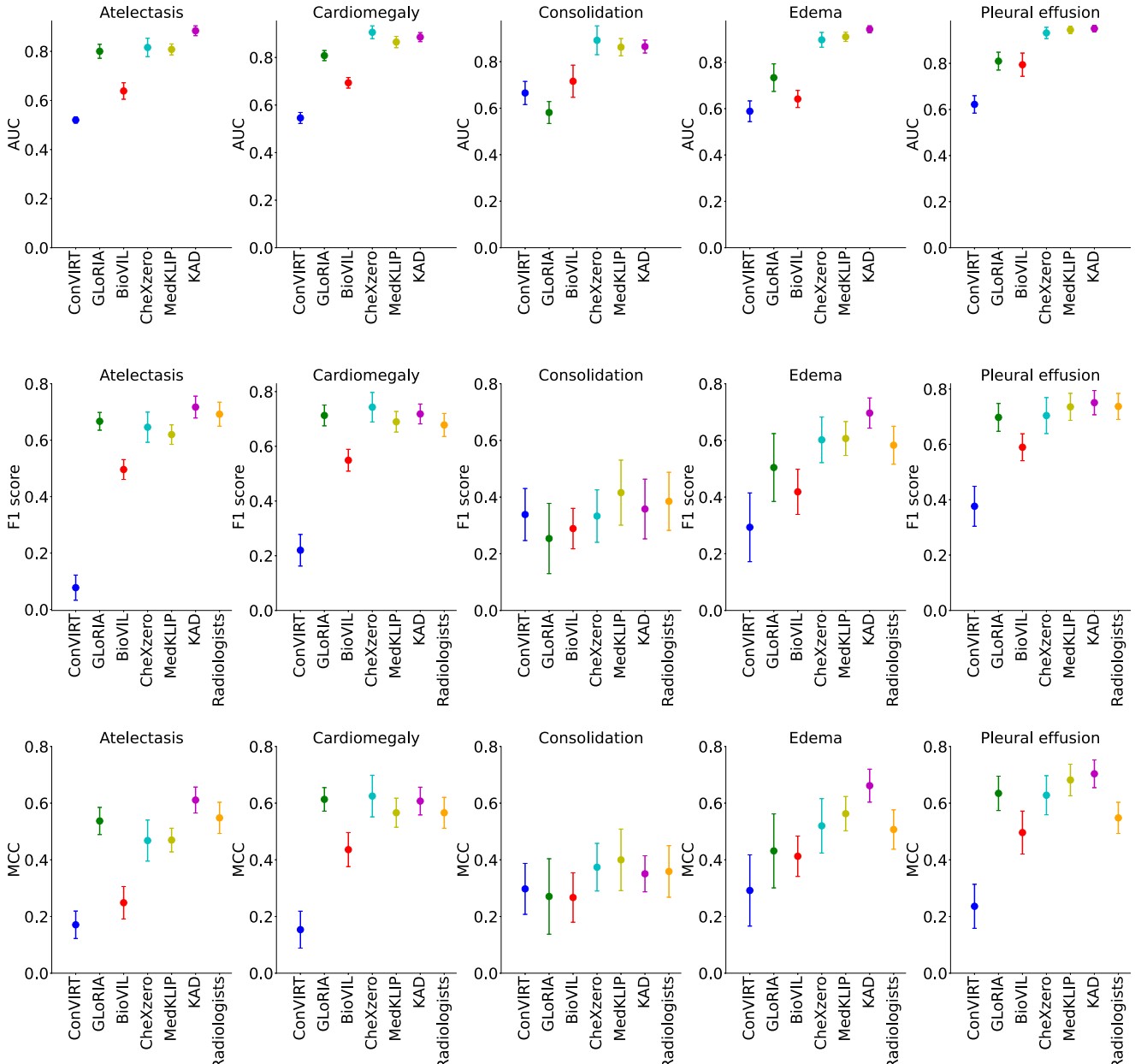

**Fig. 4 | Comparisons of proposed KAD with SOTA medical image-text pre-training models and three board-certified radiologists on five competition pathologies in the CheXpert test dataset ($n$ = 500).** Note that all models are directly evaluated on the CheXpert dataset under zero-shot setting. The AUC, F1 scores and MCC of five pathologies are shown in the plots, where the average and 95% CI are shown. Details in Supplementary Table 3.

red boxes labeled by board-certified radiologists. Quantitatively, we evaluate KAD on ChestX-Det10 and present the pointing game results in Fig. 5, which further confirms the provided explanations.

Despite the ability of KAD on zero-shot and data-efficient transfer, there remain a few limitations: First, the pre-trained model still requires a small validation set for hyper-parameter tuning, e.g., to determine the probability threshold of a specific disease. Second, some pathologies may not have any relation with other diseases in the knowledge base and never be mentioned in the reports, for example, pectum carinatum, our method is not expected to predict that pathology with high accuracy during zero-shot evaluation for this case. Third, the diagnosis of KAD is limited to classification and coarse grounding and is not able to provide accurate segmentation. In future work, we would extend the knowledge-enhanced training to more diverse medical tasks, where structured or unstructured can be acquired with negligible cost.

In summary, our proposed KAD leverages paired X-ray images and clinical reports for vision-language pre-training, incorporating medical knowledge from a well-established knowledge graph. As a result, the model has demonstrated significant performance improvement and robustness over existing approaches in auto-diagnosis for chest X-ray images. Our findings in this paper provide inspiration for resolving a few concerns that potentially hinder the feasibility of developing foundation models for more general medical diagnosis applications. First, laborious data curation. Large-scale data collection incurs time-consuming procedures, especially in unifying the data format and terminology in clinical reports. We investigate various approaches to simplify the raw reports into a set of meaningful medical terminologies, for example, using heuristic rules, RadGraph, or ChatGPT. Second, with limited and exorbitant expert knowledge, the analysis of medical image data heavily relies on domain-specific expert knowledge; KAD first learns a neural representation of the knowledge graph

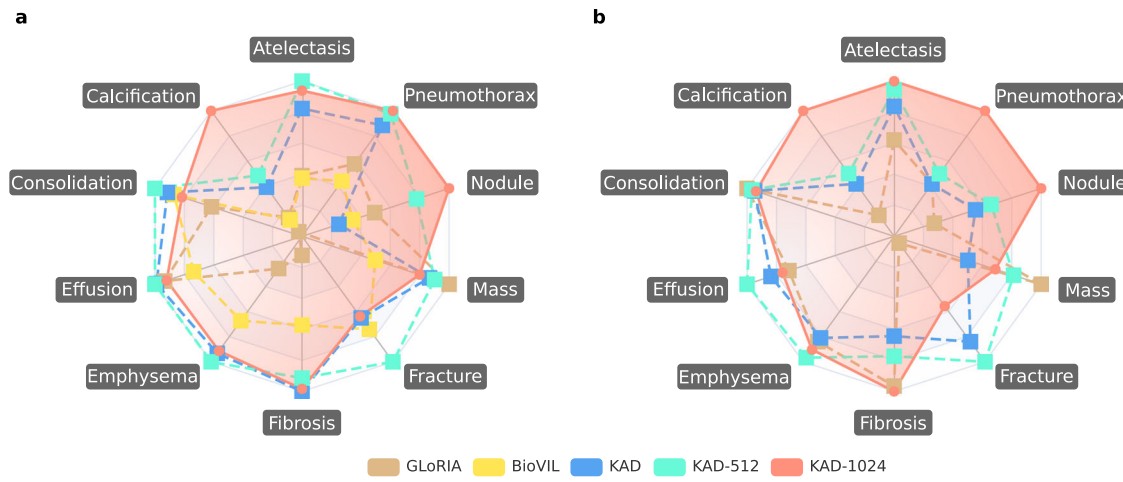

**Fig. 5 | Comparisons of proposed KAD with SOTA medical image-text pre-training models on ChestX-Det10 dataset.** AUC scores are shown for the zero-shot classification task in (**a**), and Pointing game scores are shown for the zero-shot grounding task in (**b**). We use the best results as the maximum value for each category in the radar chart, and 0.5 as the minimal value for (**a**), and 0 as the minimal value for (**b**). Details in Supplementary Table 4.

**Table 1 | Ablation study of KAD under zero-shot setting by removing or replacing individual modules**

| Methods | ChestXray14 | | | | CheXpert | | | |
|---|---|---|---|---|---|---|---|---|
| | AUC | MCC | F1 | ACC | AUC | MCC | F1 | ACC |
| Ablation on proposed modules | | | | | | | | |
| KAD w/o Stage1 | 0.752 | 0.228 | 0.274 | 0.748 | 0.894 | 0.546 | 0.620 | 0.858 |
| KAD w/o random select | 0.751 | 0.242 | 0.290 | 0.780 | 0.878 | 0.571 | 0.671 | 0.812 |
| KAD w/o DQN | 0.672 | 0.144 | 0.109 | 0.747 | 0.822 | 0.419 | 0.508 | 0.806 |
| Ablation on entity extraction module | | | | | | | | |
| w/ UMLS | 0.773 | 0.268 | 0.308 | 0.833 | 0.904 | 0.562 | 0.635 | 0.867 |
| w/ ChatGPT | 0.784 | **0.284** | **0.336** | **0.845** | 0.887 | 0.573 | 0.622 | **0.888** |
| w/ RadGraph (KAD) | **0.789** | 0.280 | 0.323 | 0.816 | **0.905** | **0.589** | **0.647** | 0.875 |

AUC, MCC, F1 and ACC scores are reported, and the metrics all refer to the macro average on all the diseases. KAD w/o Stage1 uses PubMedBERT as the text encoder. KAD w/o random select uses only image features as key or value when training DQN. KAD w/o DQN only trains the cross-modal contrastive learning part and test like previous image-text pre-training models. We also perform ablation on the entity extraction tool. w/ heuristic rule uses UMLS based on heuristic rule for entity extraction instead of RadGraph. w/ ChatGPT uses the large language model for entity extraction. w/ RadGraph utilizes the reports information extraction toolbox, RadGraph for entity extraction. Note that for fairness, all baselines use the same backbone as the basic image encoder (that is, ResNet50). Numbers within parentheses indicate 95% CI. The best results are bold.

**Table 2 | Ablation study of KAD with different image encoders and different image resolutions**

| Image Encoder | Size | ChestXray14 | | | | CheXpert | | | |
|---|---|---|---|---|---|---|---|---|---|
| | | AUC | MCC | F1 | ACC | AUC | MCC | F1 | ACC |
| ResNet-50[25] | 224 | 0.789 | 0.280 | 0.323 | 0.816 | 0.905 | 0.589 | 0.647 | 0.875 |
| ViT-16[37] | 224 | 0.785 | 0.276 | 0.321 | 0.807 | 0.907 | 0.575 | 0.647 | **0.884** |
| ResNet-50[25] | 512 | 0.788 | 0.278 | 0.321 | 0.817 | **0.908** | **0.595** | **0.656** | 0.874 |
| ResNet-50[25] | 1024 | **0.802** | **0.306** | **0.347** | **0.828** | 0.902 | 0.556 | 0.607 | 0.836 |

AUC, MCC, F1 and ACC scores are reported, and the metrics all refer to the macro average on all the diseases. Numbers within parentheses indicate 95% CI. The best results are bold.

that captures the implicit relation between medical entities in the textual embedding space, benefits the model's generalization on long-tailed target tasks. Lastly, model interpretation, KAD diagnoses with an attention map output enables the clinicians to understand the model's decision-making procedure and thus encourage trustiness.

## Methods
### Domain-specific knowledge
To incorporate the medical domain knowledge into the training procedure of our proposed Knowledge-enhanced Auto Diagnosis system, we leverage a well-established medical knowledge graph (UMLS) for pre-training and investigate various ways for medical entities extraction from reports.

**UMLS knowledge graph.** We leverage the Unified Medical Language System (UMLS)[19,23] to serve as the knowledge base for training our proposed architecture. In detail, UMLS contains medical concepts (entities) that are integrated from different lexicon resources; each entity has a Concept Unique Identifier (CUI) with corresponding definitions and multiple synonymous names and has been assigned one or occasionally multiple semantic types. The UMLS also provides relation information between medical entities in the form of triplets, normally

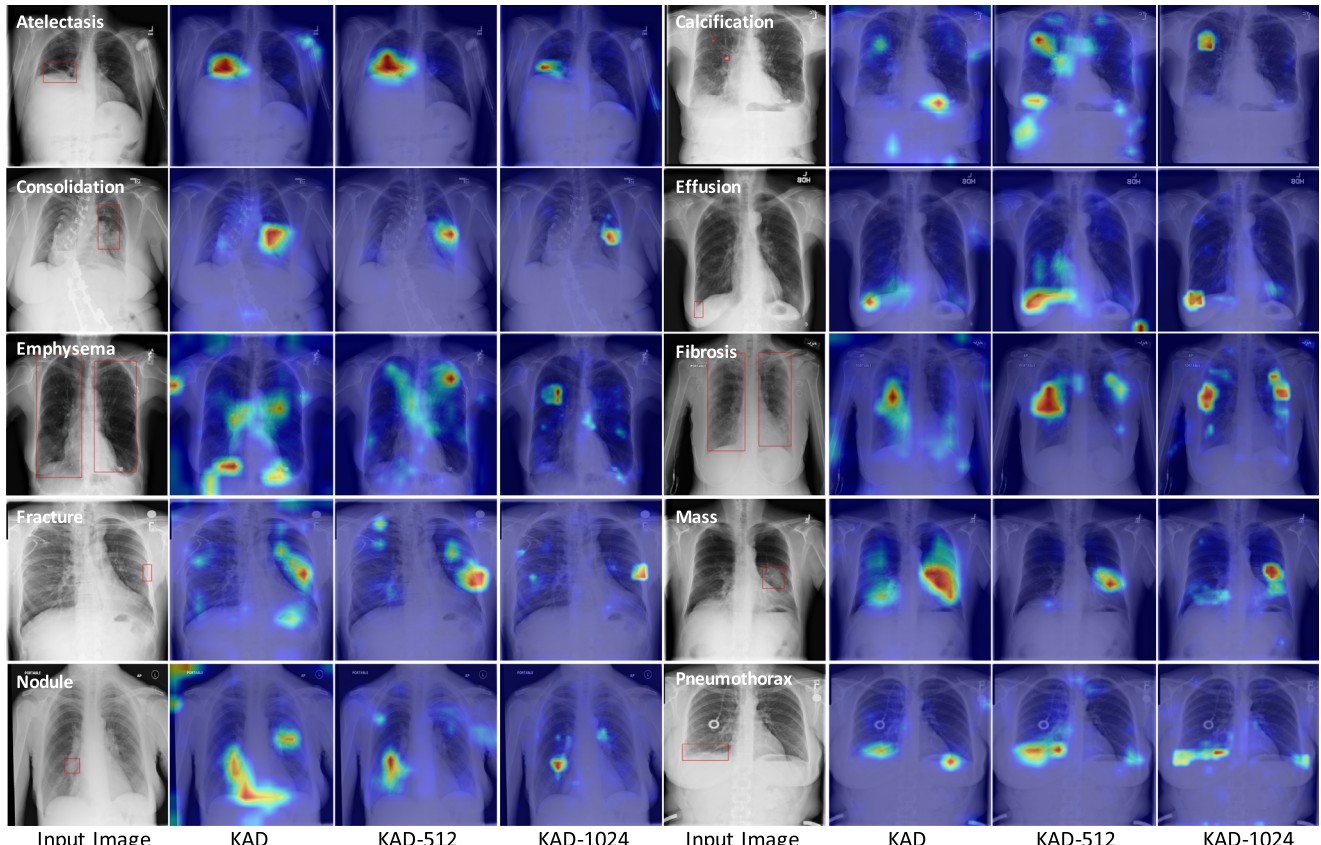

Input Image          KAD          KAD-512          KAD-1024          Input Image          KAD          KAD-512          KAD-1024

**Fig. 6 | Zero-shot visualization of randomly chosen samples from ChestX-Det10, we present both the original image (left) and attention maps generated from KAD, KAD-512, and KAD-1024.** In the original images, red boxes denote lesion areas annotated by radiologists. In the attention maps, the red to blue spectrum is plotted on the original image, with red representing high-attention regions and blue representing low attention.

represented as (head entity, relation, tail entity); for example, adenoviral pneumonia, has finding site, lungs pair, as shown in Fig. 1a.

**Entity extraction.** We explore three different ways, namely, heuristic rules, RadGraph, or ChatGPT, to preprocess the given X-ray reports, converting them from raw texts into medical entities and their presence, for example, the sentence "There is no consolidation, effusion, or pneumothorax." will be converted into {consolidation, absent, [SEP], effusion, absent, [SEP], pneumothorax, absent}. the extraction procedure will be detailed in the following sections.

### Datasets
**Dataset for pre-training.** In our experiments, we conduct model pre-training on MIMIC-CXR[14], it is a publicly available dataset of chest radiographs with radiology text reports. The MIMIC-CXR dataset contains 377,110 images corresponding to 227,835 radiographic studies for 65,379 patients. Each radiographic study comes with a free-text radiology report and corresponding chest X-ray images (in frontal or lateral views). The radiology report is a summary written by radiologists regarding their findings that consists of multiple sections: examination, indication, impression, findings, technique, and comparison. In practice, we only keep the findings and impressions sections in the reports.

**Datasets for downstream evaluation.** To evaluate the model, we conduct experiments on both zero-shot transfer and model fine-tuning. The details of the datasets and implementation are described below.

PadChest[15]. PadChest has 160,868 chest X-ray images labeled with 174 different radiographic findings, 19 differential diagnoses; only 27% of the labels (totaling 39,053 examples) come from board-certified radiologists, and the rest are obtained by using a recurrent neural network with attention trained on the radiology reports. For evaluation purposes, we only test on samples annotated by board-certified radiologists and report the zeros-shot transfer results.

ChestXray14[16]. NIH ChestXray14 has 112,120 chest X-ray images with disease labels from 30,805 unique patients. The disease labels are acquired by mining the associated radiological reports with natural language processing tools. In total, there are 14 disease labels: Atelectasis, Cardiomegaly, Effusion, Infiltration, Mass, Nodule, Pneumonia, Pneumothorax, Consolidation, Edema, Emphysema, Fibrosis, Pleural Thickening and Hernia. In this paper, all models are tasked to predict a binary label for each of these 14 disease labels for each X-ray image from the dataset, resembling a multi-label classification problem. We strictly follow the official patient-wise data split from the original ChestXray14 release and use the images from 10% of subjects in the training set to form a validation set.

CheXpert[17]. CheXpert has 224,316 chest X-ray images collected from 65,240 patients. The official validation set contains 200 chest radiographic studies that are manually annotated by three board-certified radiologists, and the official test set contains 500 chest radiographic studies annotated by a consensus of five board-certified radiologists[24]. We use the validation dataset for picking the thresholds of predictions. We follow the original paper and focus on the evaluation of five observations on the official test set (the competition tasks): Atelectasis, Cardiomegaly, Consolidation, Edema, and Pleural Effusion.

ChestX-Det10[18]. ChestX-Det10 is a subset of NIH ChestXray14, which consists of 3543 chest X-ray images with box-level annotations provided by three board-certified radiologists of 10 diseases/abnormalities, including Atelectasis, Calcification, Consolidation,

Effusion, Emphysema, Fibrosis, Fracture, Mass, Nodule and Pneumothorax. In this paper, we follow the official data split and report the zero-shot grounding results on the test set.

## Baselines

In general, existing state-of-the-art models can be cast into two categories based on the modalities involved at the pre-training stage: medical image-text pre-training approaches, which enable both zero-shot transfer and fine-tuning, medical image pre-training approaches, which only support fine-tuning evaluation. Note that all the above baselines are implemented using the same backbone architecture as our KAD, i.e., a ResNet-50[25] architecture as the basic image encoder module. Such an implementation decision rules out the effect of network architectures on final performance and advocates fairness in experimental comparisons.

**Medical image-text pre-training methods.**
- ConVIRT[10] jointly trains the vision and text encoders with the paired medical images and reports via bidirectional contrastive learning.
- GLoRIA[9] models the interactions between medical images and reports via both global and regional contrastive learning.
- BioVIL[26] leverages a pre-trained radiology-specific text encoder for contrastive learning using paired image-text radiology data.
- CheXzero[22] fine-tunes the pre-trained CLIP model[4] on the X-ray and reports with contrastive learning using image-text pairs.
- MedKLIP[27] leverages medical-specific knowledge descriptions to enhance visual-language pre-training using paired image-text data.

**Medical image pre-training methods.**
- Model Genesis[28] refers to the self-supervised visual representation learning approach that considers restoring the original image from artificial distortion, including local shuffling, non-linear transformation, in- and out-painting.
- Comparing to Learn (C2L)[29] refers to the self-supervised visual representation learning approach that generates positive and negative pairs through both image and feature-level mix-up for contrastive learning.
- ImageNet Pre-training[30] refers to a representative method that is pre-trained on a large-scale dataset of natural images with supervised learning.

## Data preprocessing

At the visual-language pre-training stage, we resize each chest X-ray image to 224 × 224 by default, then normalize with mean and standard deviation computed from the whole training dataset. Random resize crop, random horizontal flip, random rotation (−10 to 10 degrees), and random grayscale (brightness, sharpness, and contrast) are also applied for augmentation. For the text from radiology reports, we first apply the named entity recognition and linking tool, ScispaCy[31], to preprocess the texts, i.e., extract the entities from reports and ground them in the UMLS knowledge base for entity disambiguation. Then, we use RadGraph[20] to obtain the corresponding semantic types for each entity with uncertainty levels. The entities with corresponding semantic types and uncertainty levels are concatenated as input, e.g., {pleural effusion, observation definitely present, [SEP], lung, anatomy, [SEP],...}, and the maximal sequence length is 256. At the fine-tuning stage, we apply the same set of strategies for preprocessing as in pre-training for all four target domain datasets.

## Algorithm overview

At the core of our proposed idea is to train foundation models by leveraging the medical prior knowledge; in this paper, we explicitly leverage a well-established medical knowledge graph, i.e., Unified Medical Language System (UMLS), and various information extraction

methods based on heuristic rules, RadGraph, or ChatGPT. Generally speaking, we train the model in two stages; first, we train a knowledge encoder to learn a neural representation of the medical knowledge base; second, we extract the clinical entities and relations from the radiology reports and employ the pre-trained knowledge encoder to guide visual representation learning on image and text pairs of chest X-rays. In the following sections, we start by describing our considered problem scenario and then present the detail of these two stages sequentially.

**Problem scenario.** Assuming a training set with $N$ samples, i.e., $\mathcal{D}_{\text{train}} = \{(x_1, t_1), \ldots, (x_N, t_N)\}$, where $x_i, t_i$ refer to the X-ray image and its corresponding medical report, respectively, our goal is to train a visual-language model that enables the diagnosis of the existence of any pathologies. Specifically, at inference time, we can freely ask the system to identify the likelihood of the patient getting a certain disease:

$$\hat{s}_i = \Phi_{\text{DQN}}(\Phi_{\text{image}}(x_i), \Phi_{\text{knowledge}}([\text{disease}])), \tag{1}$$

where $x_i \in \mathbb{R}^{H \times W \times 3}$ refers to an image sample from the test set, with $H, W$ denoting height and width, respectively, $\hat{s}_i \in [0,1]$ refers to the inferred likelihood of the patient having a certain disease.

**Knowledge encoder.** In this section, we provide details for training the knowledge encoder ($\Phi_{\text{knowledge}}$), with experts' knowledge, to implicitly model the relations between medical entities in textural embedding space. Specifically, we employ contrastive learning to fine-tune a pre-trained BERT by sampling positives and negatives from Unified Medical Language System (UMLS)[19].

Let $\mathcal{D}_{\text{UMLS}} = \{(n, c, d)_i\}_{i=1}^{\|D\|}$ denote a concept dictionary for UMLS; each concept ($n_i$) has a Concept Unique Identifier (CUI, $c_i$) with corresponding definition ($d_i$) and Type Unique Identifier (TUI), as shown in Fig. 1. The concept "pulmonary infiltrate" is defined as "A finding indicating the presence of an inflammatory or neoplastic cellular infiltrate in the lung parenchyma", and the corresponding CUI is C0235896. For the UMLS knowledge graph, the vertices are concepts ($\mathcal{D}_{\text{UMLS}}$), and each edge can be represented as a triplet, i.e., $(n_i, r, n_j)$, where $n_i$ refers to the head concept, $n_j$ refers to the tail concept, and $r$ refers to the relation from the head concept to the tail concept.

Here, we train the knowledge encoder by maximizing the similarities between positive concept-definition pairs and concept-relation-concept triplets generated from the knowledge graph, such that the language description pointed to the same CUI has similar representations.

For a specific CUI $c_i$, the corresponding language descriptions may be expressed in three formats: the concept $n_i$, the definition $d_i$, and other concepts with the correct relationship $\{n_j + r\}$, $r$ is the relationship between the head concept $n_j$ pointing to tail concept $n_i$. Note that there may exist more than one head concept pointing to the same tail concept, and we randomly select one of them for training. Given randomly $N$ sampled CUIs, we can compute the textual embedding of corresponding language descriptions, i.e., the concept $\boldsymbol{n}_i \in \mathbb{R}^{N \times d}$, definition $\boldsymbol{d}_i \in \mathbb{R}^{N \times d}$, and concepts with the relationship $\{\boldsymbol{n}_j + r\} \in \mathbb{R}^{N \times d}$. Each pair of textual embeddings with the same CUI can be treated as positive, and the ones with different CUIs as negative pair.

At training time, for each CUI, we randomly sample two textual embeddings accordingly, and each mini-batch can be expressed as $\{(z_i, c_i)\}_{i=1}^{2N}$, where $c_i$ denotes the CUI pointed by $z_i$. Here, we unify the textual embedding after knowledge encoder $\Phi_{\text{knowledge}}$ as $z$. Therefore, the model can be trained via contrastive learning by minimizing the

distances between the $z_i$ that points to the same CUI:

$$\mathcal{L}_{\text{knowledge}} = \sum_{i=1}^{2N} \frac{-1}{2N_{c_i}-1} \sum_{j=1}^{2N} \mathbb{1}_{i \neq j} \cdot \mathbb{1}_{c_i = c_j} \cdot \log \frac{\exp\left(\boldsymbol{z}_i \cdot \boldsymbol{z}_j / \tau\right)}{\sum_{k=1}^{2N} \mathbb{1}_{i \neq k} \cdot \exp\left(\boldsymbol{z}_i \cdot \boldsymbol{z}_k / \tau\right)}, \tag{2}$$

where $N_{c_i}$ denotes the number of $\boldsymbol{z}_i$ with label $c_i$ in this minibatch, $\tau \in \mathbb{R}^+$ is a scalar temperature parameter, and $\mathbb{1}$ is the indicator function.

**Entity extraction.** In this section, we introduce the entity extraction module for processing text reports, into medical entities, for example, anatomy or observation, with their presence information. Anatomy refers to an anatomical body part that occurs in a radiology report, such as a "lung". Observation refers to the word associated with visual features, physiological processes or diagnostic disease classifications, such as "effusion" for each observation, the uncertainty level of its existence will be specified as present or absent. For each radiology text with more than one sentence, they will be converted into a sequence of entities and categories, i.e., $t = \{e_1^1, s_1^1, [\text{SEP}], \ldots e_i^k, s_i^k, [\text{SEP}] \ldots\}$, where $e_i^k$ is the $k$-$th$ extracted entity for $i$-$th$ sentence, $s_i^k$ is the category of $e_i^k$. The [SEP] token separates the entities. After extracting all entities from the reports, we select the top $Q$ most commonly appearing observation entities from the entire reports corpus, denoted as an entity set $\mathcal{Q} = \{q_1, q_2, \ldots, q_Q\}$, and get a label indicating the existence of the entity from the uncertainty level. For entities that are not mentioned in the report, we set their label as "absent" by default. We make three attempts at this goal.

First, we employed heuristic rules using the Unified Medical Language System (UMLS). Specifically, for each sentence in the radiology report, we were able to extract a sequence of entities (entity, concept, CUI, TUI) with the Python `spacy` package. The pseudo-code is provided in Supplementary Fig. 6. To determine the presence of an entity, we first filter the entity list for each sentence based on TUI, retaining only the entities with TUI in (T033: Finding, and T047: Disease or Syndrome). Next, we match these entities with our entity set, except for 'normal'. Furthermore, we adopted a straightforward heuristic rule: if the sentence contains the words 'no', 'none', or 'normal', the label is set to absent; otherwise, the label is set to 1 present. For entities that are not mentioned in the report, we assigned a label of absent. If all entities are absent, we set the label of 'normal' to present.

Second, we utilized the off-the-shelf RadGraph, for identifying radiology-specific entities and asserting their presence and anatomical relations from clinical text. RadGraph was trained on 500 radiology reports from the MIMIC-CXR dataset annotated by board-certified radiologists. With RadGraph, each radiology text was directly converted into a sequence of entities and categories. Each entity $e_i^k$ will be classified into one of the following categories, defined as $s_i^k$: Anatomy (ANAT), Observation: Definitely Present (OBS-DP), Observation: Uncertain (OBS-U), and Observation: Definitely Absent (OBS-DA). For entities that are not mentioned in the report, we set their label as "definitely absent" by default.

Third, we experiment with ChatGPT[32], a large language model that has demonstrated remarkable results in natural language processing. We input the radiology report as the content to ChatGPT and use the following prompt:

> For the given report of a chest x-ray, determine if the patient has any of the following findings: finding_list = [pleural effusion, opacity, pneumothorax, edema, atelectasis, tube, consolidation, enlarged cardiomediastinum, tip, pneumonia, line, cardiomegaly, fracture, calcification, medical device, engorgement, nodule, wire, pacemaker, pleural thicken, marking, scar, hyperinflate, blunt, collapse, emphysema, aerate, mass, infiltration, obscure, deformity, hernia, drainage, distention, shift, lesion, hardware, dilation, aspiration]. The output

should use the following template: label_list = [i if finding exists for finding in finding_list]. If no finding exists, output label_list = []

We then processed the output label_list to obtain the presence of an entity.

**Knowledge-guided visual representation learning.** This section describes the detail of knowledge-guided visual representation learning. In particular, we introduce individual components of our architecture, including image encoder ($\Phi_{\text{image}}$), knowledge encoder ($\Phi_{\text{knowledge}}$), disease query network ($\Phi_{\text{dqn}}$). Lastly, we will describe the training procedure.

Image encoder. Given an X-ray image scan $x_i \in \mathbb{R}^{H \times W \times 3}$, we compute the features with a visual backbone:

$$\boldsymbol{x}_i = \Phi_{\text{image}}(x_i) \in \mathbb{R}^{m_x \times d}, \tag{3}$$

where $d$ refers to the feature dimension, and $m_x = h \times w$ with $h, w$ denoting the size of the output feature map. In our case, if the standard ResNet-50[25] is adopted as the visual backbone, we take the output from the fourth residual block. And if the standard ViT-16 is used, we simply use the features from the transform encoder output.

Knowledge encoder. Given a preprocessed text report $t_i$, we compute the features with the pre-trained knowledge encoder:

$$\boldsymbol{t}_i = \Phi_{\text{knowledge}}(t_i) \in \mathbb{R}^{m_t \times d}, \tag{4}$$

where $d$ refers to the feature dimension, and $m_t$ refers to the token number.

Disease query network. Given the pre-defined entity set $\mathcal{Q}$, we compute a set of query vectors with the pre-trained knowledge encoder, $\mathcal{Q} = \{\boldsymbol{q}_1, \boldsymbol{q}_2, \ldots, \boldsymbol{q}_Q\}$, where $\boldsymbol{q}_i = \Phi_{\text{knowledge}}(q_i)$. At training time, we randomly pick the encoded visual features $\boldsymbol{x}_i$ or text features $\boldsymbol{t}_i$ to act as the key and value of the disease query network ($\Phi_{\text{dqn}}$), corresponding to the "Random Select" module in Fig. 1c. As ideally, we would like the visual and textual embedding space to be used interchangeably.

$$\hat{s}_i = \Phi_{\text{dqn}}(\boldsymbol{x}_i, \boldsymbol{t}_i, \mathcal{Q}) \in \mathbb{R}^{Q \times d}. \tag{5}$$

Note that during inference, we only use the visual features as the input of DQN. The outputs from the DQN are further fed into an MLP for inferring the existence of the query entity.

Training. We randomly sample a minibatch of N input pairs from the training data and optimize the contrastive loss:

$$\mathcal{L}_{\text{contrast}} = -\left( \log \frac{e^{(\langle \hat{\boldsymbol{x}}_i, \hat{\boldsymbol{t}}_i \rangle / \tau)}}{\sum_{k=1}^{N} e^{(\langle \hat{\boldsymbol{x}}_i, \hat{\boldsymbol{t}}_k \rangle / \tau)}} + \log \frac{e^{(\langle \hat{\boldsymbol{t}}_i, \hat{\boldsymbol{x}}_i \rangle / \tau)}}{\sum_{k=1}^{N} e^{(\langle \hat{\boldsymbol{t}}_i, \hat{\boldsymbol{x}}_k \rangle / \tau)}} \right). \tag{6}$$

$\mathcal{L}_{\text{contrast}}$ denotes an image-to-text and text-to-image contrastive loss for the $i$-th pair, respectively, where $\hat{\boldsymbol{x}}_i$ denotes the mean pooling over visual feature $\boldsymbol{x}_i$, $\hat{\boldsymbol{t}}_i$ denotes the mean pooling over text feature $\boldsymbol{t}_i$, $\langle \cdot, \cdot \rangle$ represents the cosine similarity, $\tau$ represents a temperature parameter.

As aforementioned, for each sample, we can obtain the existence label of the entity set $\mathcal{Q}$ with the entity extraction module. Then for existence prediction, we use binary cross-entropy with the existence label, denoted as $\mathcal{L}_{\text{dqn}}$. Finally, for each mini-batch, we simply sum up $\mathcal{L}_{contrast}$ and $\mathcal{L}_{dqn}$ as the overall loss.

$$\mathcal{L}_{\text{KAD}} = \mathcal{L}_{\text{contrast}} + \mathcal{L}_{\text{dqn}} \tag{7}$$

**Implementation details**

Model pre-training. The overall framework generally follows a two-stage training pipeline. For Stage1, we initialize the text encoder from the English version PubMedBERT[33] and fine-tune it for 100K

training steps. In each mini-batch, 64 concept-definition pairs and 64 concept-relation-concept triplets are used for training. The maximal sequence length is 256 since the definition could be long. We use AdamW[34] as the optimizer with $lr = 1 \times 10^{-4}$ and $lr_{warmup} = 1 \times 10^{-5}$. For Stage2, the image encoder is the first four layers of ResNet50, the DQN is composed of a series of standard Transformer decoders[35]. We use AdamW optimizer with $lr = 1 \times 10^{-4}$ and $lr_{warmup} = 1 \times 10^{-5}$. We train on an A100 with batch size 64 for 50 epochs. The first five epochs are set for warming up.

Zero-shot transfer. To evaluate the zero-shot performance of the model on the multi-label classification task, DQN takes the disease name list as query input and image features as key and value and output the likelihood of the disease being present in the considered chest X-ray image. The average cross-attention between the disease and the visual features is used for grounding. For other medical image-text pre-training baselines, we use the prompt defined in BioVIL[26], for example, representing the presence/absence of pneumonia: "Findings suggesting pneumonia" and "No evidence of pneumonia".

Model fine-tuning. At this stage, we fine-tune all models using AdamW optimizer with $lr = 1 \times 10^{-4}$ for all datasets. The model is trained with the same learning rate and decay strategy as that used in the pre-training stage and is trained with a batch size of 64. All experiments were run in Python 3.9.12 and torch '1.9.1+cu111'.

## Ablation study

We conduct a thorough ablation study of KAD, systematically varying one variable at a time. By default, our model comprises the image encoder, knowledge encoder, random select module, and disease query network and incorporates RadGraph for entity extraction. Table 1 shows the quantitative results, indicating that the proposed modules are effective in addressing the limitations of the previous approach: knowledge deficit, report complexity, and limited transferability. We also perform experiments to analyze the impact of image resolution and visual backbone (Table 2).

Knowledge encoder. "KAD w/o Stage1" refers to using PubMedBERT[33] as text encoder without fine-tuning on medical knowledge graph. We show the similarity map between features of different disease names encoded by PubMedBERT and the pre-trained text encoder in Supplementary Fig. 7; our knowledge-enhanced text encoder clearly distinguishes the features of different diseases, while PubMedBERT encodes them almost identically.

Cross-modal contrastive learning without entity extraction. "KAD w/o entity extraction" refers to not using entity extraction module in cross-modal contrastive loss (Eq. (6)), i.e., simply aligning the raw reports and images. As shown in the table, with the entity extraction module, the classification results can be improved from 0.772 to 0.786 on AUC in ChestX-ray14 dataset and 0.897 to 0.906 on AUC in CheXpert dataset. These results demonstrate the benefits of standardized reports for training.

Methods for entity extraction. "w/ UMLS" refers to employing only heuristic rules for entity extraction. "w/ RadGraph", "w/ ChatGPT" refers to utilizing RadGraph and or ChatGPT, respectively. Based on the results, it can be observed that although heuristic rules slightly underperform in comparison, it is already very competitive, and RadGraph and ChatGPT perform equally well. It should be noted that different approaches for extracting entities yielded similar results, indicating that the specific method used for entity extraction was not the primary factor affecting performance in KAD. The critical component is our proposed DQN architecture.

Random select module. "KAD w/o random select" refers to removing the random select module, i.e., only using the image features as the key and value of the disease query network. The results in Table 1 verify that the random select module help to align image features and

text features in the textual embedding space, thus improving the zero-shot performance.

Disease query network. "KAD w/o DQN" refers to only training with contrastive learning in Stage2, i.e., similar design as the existing image-text pre-training models, which incurs the most significant performance degradation, showing the necessity of DQN for better generalization on disease diagnosis or radiology findings.

Image encoders and image resolutions. As shown, performance with ResNet and ViT are comparable, demonstrating that the proposed knowledge-enhanced representation learning is robust to the selection of image encoder. And the results of KAD-512 and KAD-1024 show that the ability to diagnose is not noticeably improved with the image resolution increasing, compared to image localization (Fig. 5).

## Statistical analysis

In our evaluation, AUC stands for "Area under the ROC Curve", MCC stands for "Matthews Correlation Coefficient", F1 stands for "F1 score" and ACC stands for "Accuracy". We collect AUC results directly using the model's prediction, while to obtain the MCC, we first run inference on the test set to get the probability values for the different classes on each chest X-ray image. The probabilities are then transformed into positive/negative predictions with thresholds found by optimizing MCC over the validation dataset. Then, the condition-based MCC scores are calculated using these predictions. We similarly compute the F1 score and ACC but use the same thresholds as used for computing the MCC. For zero-shot grounding, we use Pointing Game[36] for evaluation purposes. In specific, we extract the region with maximum response in the output heatmap, and if the region hits the ground-truth mask, it is considered a positive prediction, otherwise negative. Finally, accuracy can be calculated as the pointing game score.

## Confidence intervals

We use the non-parametric bootstrap to generate confidence intervals: random samples of size $n$ (equal to the size of the original dataset) are repeatedly sampled 1000 times from the original dataset with replacement. We then estimate the AUC, MCC, F1 and ACC metrics using each bootstrap sample. The predicted probabilities are then transformed into positive/negative predictions using the thresholds found by optimizing MCC over the validation dataset. We derive confidence intervals from the relative frequency distribution of the estimates over the re-samples, using the interval between the $100 \times (\alpha/2)$ and $100 \times (1 - \alpha/2)$ percentiles; we pick $\alpha = 0.05$.

## Reporting summary

Further information on research design is available in the Nature Portfolio Reporting Summary linked to this article.

## Data availability

MIMIC-CXR data is available at https://physionet.org/content/mimic-cxr/2.0.0 for users with credentialed access. PadChest data is available at https://bimcv.cipf.es/bimcv-projects/padchest. NIH ChestXray14 data is available at https://nihcc.app.box.com/v/ChestXray-NIHCC/folder/36938765345. CheXpert data is available at https://aimi.stanford.edu/chexpert-chest-x-rays, and the official test data with labels are available at https://github.com/rajpurkarlab/cheXpert-test-set-labels. ChestX-Det10 data are available at https://github.com/Deepwise-AILab/ChestX-Det10-Dataset. Source data for figures are provided with this paper. Source data are provided with this paper.

## Code availability

The code is available on GitHub at https://github.com/xiaoman-zhang/KAD.

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

## Acknowledgements

The work is supported by the National Key R&D Program of China (No. 2022ZD0160702), STCSM (No. 22511106101, No. 18DZ2270700, No. 21DZ1100100), 111 plan (No. BP0719010), and State Key Laboratory of UHD Video and Audio Production and Presentation.

## Author contributions

X.Z., C.W., and W.X. conceptualized and led the research project. X.Z. wrote the code, performed the experiments, and plotted the figures. X.Z., C.W. and W.X. analyzed the results and drafted the manuscript. W.X. and Y.Z. carried out critical revisions of the manuscript and results discussion. W.X. and Y.W. supervised the projects, approved

the submission and accepted responsibility for the overall integrity of the paper.

## Competing interests

The authors declare no competing interests.
