## [Peer Review File · Nature Communications]

Knowledge-enhanced Visual-Language Pre-training on Chest Radiology ImagesEditorial Note: Parts of this Peer Review File have been amended to remove third-party material where no permission to publish could be obtained.

REVIEWER COMMENTS

Reviewer #1 (Remarks to the Author):

This paper proposed a knowledge-enhanced vision-language pre-training approach for auto-diagnosis on chest X-ray images. The prior medical knowledge is learned with a text encoder through contrastive learning based on a knowledge graph. The disease query network (DQN) is then pretrained with X-ray images and the corresponding reports. The pre-trained network can be exploited in a zero-shot inference scenario for disease diagnosis based solely on an X-ray image. The manuscript is overall organized and presented well. The proposed method is novelty. However, I have a few major concerns, especially on comparison experiments. Detailed comments are as follows:

1. Overall, the proposed method is presented clearly. However, a major component in the pretraining stage is not described at all in the text. In Fig. 1c, there is a gray box labeled as "Random select." The function of this module is never explained. This module is important for a reader to understand how the image and report are leveraged during pre-training. In my understanding, this module randomly selects either the image branch or the text branch for the following DQN.

2. The zero-shot inference capability of the proposed method is touted as the major contribution of this work, compared with the previous fully supervised approaches. However, there are no explicit evaluations of this capability. The proposed method is pre-trained on MIMIC-CXR with 19 diseases. Therefore, for these 19 diseases, the proposed method is a fully supervised model; for other diseases, it can be regarded as a zero-shot model. However, zero-shot inference capability is not explicitly evaluated. For example, for the results on the PadChest dataset as shown in Table 1, the performance on seven diseases (plus normal) was reported for both supervised and zero-shot models. It is not clear which diseases are never seen by the proposed method. Since these seven diseases are quite common, I guess that most diseases are seen. If this is the case, the performance improvement against CheXNet, a supervised model, is not impressive: for most diseases the improvement is minor and it is significantly worse than CheXNet on "lesion." The same concern applies to the other two datasets. Please explicitly evaluate the proposed method on seen and unseen diseases separately.

3. The comparison with truly zero-shot models is unfair. As said before, the proposed method is not a truly zero-shot model. First, it is specifically designed for the disease classification task; therefore, it cannot be extended to other tasks, e.g., lesion detection and segmentation. The other zero-shot models compared in Table 1 are general model, not specifically designed for any task; therefore, they may be more versatile. Second, as said above, the authors reported the performance of the proposed model on both seen and unseen diseases, without clearly indicating which diseases are seen and which are unseen. Therefore, the comparison is unfair. The authors just limit the comparison to unseen diseases for a fair comparison with those zero-shot models.

4. The authors only evaluated with the ResNet50 backbone as the image encoder, which is a quite old model. Recently, various Transformer models have been proposed and achieved superiority performance. The authors are encouraged to evaluate with more-recent backbones to verify if the findings of this work still hold.

Reviewer #2 (Remarks to the Author):

Overall this is a sufficiently interesting article on tackling chest x-ray image based disease classification using knowledge-graph structured pre-training and image-report pairs based pre-

training. The proposed system by authors' report achieved very appealing results under zero-shot learning and a few-shot tuning. Experiments have been conducted relatively extensively on three public datasets and a number of competing methods are compared.

Let me jump the critical questions that I have here.

1, The deep learning model takes resized 224x224 pixel images as inputs and everything were learned from this resolution. The image resolution of 224x224 was adopted because of the original imageNet images? Even using ResNet-50, only the convolutional layers are usable anyway so any image resolution should be able to be handled. The FC or other decision layers need to be re-trained anyway. At 224x224, each pixel is about 2mm to 3mm size (the original X-ray pixel size is about 0.1mm). Patients take 3000x4000 x-ray images for a valid reason and many diseases (especially subtle ones) can be seen clearly if the image size is smaller than ~1000 pixels (see the ChestXray14 CVPR 2017 paper). This chest xray screening population are expected to have significant appearance of spatially subtle diseases. Author should investigate using 1024x1024 image resolutions. 224x224 has essential limitations which should be obvious.

2, Authors should perform more rigorous experiments on localization of the diseases. As now, there is only one figure with a few examples. This is not enough. If the AI cannot localize the disease, it can not be trusted for its diagnosis.

<https://arxiv.org/pdf/2010.10298.pdf>

<https://arxiv.org/abs/2006.10550>

The above dataset can serve the purpose of localization.

3, The AI system has high or higher false positive rates on classifying healthy patients? The majority population in a clinical setting where chest X-ray exams are performed may be healthy patients

Reviewer #3 (Remarks to the Author):

This paper proposed to incorporate knowledge about biomedical entities and relations in vision-language pretraining using MIMIC-CXR and evaluated on zero-shot classification of Chest X-ray images. The authors reported results that outperformed other vision-language foundation models and matched supervised and average expert results.

Biomedical vision-language pretraining is an exciting research frontier and incorporating free-available knowledge to aid pretraining is a good direction to explore. However, there are key questions about the proposed methodology that make assessing its significance challenging.

Instead of pretraining on raw report text, the proposed method used radiology findings produced by RadGraph (5.4.3 "entity extraction"). This is problematic as RadGraph is a supervised method trained on reports annotated with common findings by board-certified radiologists. So effectively, the proposed method was pretrained with findings extracted by a supervised method, which made the comparison with other truly self-supervised approaches (pretrained on image-report pairs only) not very meaningful. This is the most serious concern about the proposed method, which undermines the central claim that the superiority in the proposed pretraining stems from incorporating domain knowledge. To evaluate truly self-supervised performance, the authors should not use RadGraph or any supervised entity extraction systems at all, but instead start from raw report text.

Details about knowledge-enhanced pretraining are also a bit unclear. I have a hard time figuring out what DQN does in pretraining. By definition, there is no annotated disease for an image-report pair. Where would the ground-truth label come from to compute cross entropy loss?

Eq 6 is also a bit confusing. The contrastive loss should try to pull corresponding pair closer so I

expect x_i and t_i in the numerator. Instead, the equation has x_i/t_j , t_i/x_j . I don't see any explanation about j and relation between i and j .

Respond to Review: NCOMMS-23-03142-T

Knowledge-enhanced **Visual-Language** Pre-training on Chest **Radiology** Images

Anonymous submission

We thank all reviewers for their valuable comments. In the following, we aim to resolve their concerns and will update the final manuscript accordingly. All codes and models will be released.

Reviewer 1

Q 1.1 Overall, the proposed method is presented clearly. However, a major component in the pretraining stage is not described at all in the text. In Fig. 1c, there is a gray box labeled as “Random select”. The function of this module is never explained. This module is important for a reader to understand how the image and report are leveraged during pre-training. In my understanding, this module randomly selects either the image branch or the text branch for the following DQN.

Reply: Yes. With the “random select” module, the model will randomly pick the encoded visual features \mathbf{x}_i or text features \mathbf{t}_i as input to the following DQN. We have added the description of this module in Section 5.4.3, and the ablation study (Respond Letter Table 1) of this module. The revision is shown as follows.

“At training time, we randomly pick the encoded visual features \mathbf{x}_i or text features \mathbf{t}_i to act as the key and value of the disease query network (Φ_{dqn}), corresponding to the “Random Select” module in Fig. 1c. As ideally, we would like the visual and textual embedding space to be used interchangeably.”

Methods	ChestXray14				CheXpert			
	AUC	MCC	F1	ACC	AUC	MCC	F1	ACC
KAD w/o random select	0.751	0.242	0.290	0.780	0.878	0.571	0.671	0.812
KAD	0.789	0.280	0.323	0.816	0.905	0.589	0.647	0.875

Respond Letter Table 1: Ablation study of KAD under **zero-shot** setting. KAD w/o random select uses only image features as key or value when training DQN. AUC, MCC, F1 and ACC scores are reported, and the metrics all refer to the macro average on all the diseases.

Q 1.2 The zero-shot inference capability of the proposed method is touted as the major contribution of this work, compared with the previous fully supervised approaches. However, there are no explicit evaluations of this capability. The proposed method is pre-trained on MIMIC-CXR with 19 diseases. Therefore, for these 19 diseases, the proposed method is a fully supervised model; for other diseases, it can be regarded as a zero-shot model. However, zero-shot inference capability is not explicitly evaluated. For example, for the results on the PadChest dataset as shown in Table 1, the performance on seven diseases (plus normal) was reported for both supervised and zero-shot models. It is not clear which diseases are never seen by the proposed method. Since

these seven diseases are quite common, I guess that most diseases are seen. If this is the case, the performance improvement against CheXNet, a supervised model, is not impressive: for most diseases the improvement is minor and it is significantly worse than CheXNet on “lesion.” The same concern applies to the other two datasets. Please explicitly evaluate the proposed method on seen and unseen diseases separately.

Reply: Thanks for the suggestion. We have revised the paper to report the performance on seen or unseen classes separately, we here give a brief summary.

Zero-shot inference means the model requires zero training while shifting to a new dataset at inference time. In this paper, our KAD model is pre-trained on MIMIC-CXR, to distinguish the seen and unseen diseases, we denote the diseases that appear in the “entity set” (obtained from MIMIC-CXR) as seen diseases, and the others as unseen diseases. While evaluating on the following three datasets, the detailed data division is as follows :

- **CheXpert:** collected from Stanford Hospital, all 5 classes are seen at training time;
- **ChestXray14:** collected from NIH Clinical Center, with 15 seen classes (14 seen and only 1 unseen, fibrosis);
- **PadChest:** collected from Indiana University, with a total of 193 classes (16 seen and 177 unseen).

As shown in Respond Letter Table 2, while comparing KAD with existing SOTA self-supervised approaches on three datasets, KAD maintains consistent superiority, especially for unseen classes.

Methods	CheXpert	ChestXray14		PadChest	
	Seen classes	Seen classes	Unseen classes	Seen classes	Unseen classes
ConVIRT [9]	0.591	0.566	0.482	0.523	0.515
GloRIA [3]	0.750	0.621	0.459	0.492	0.487
BioVIL [1]	0.694	0.665	0.632	0.708	0.645
CheXzero [6]	0.889	0.694	0.598	0.758	0.666
MedKLIP [7]	0.879	0.735	0.604	0.668	0.636
KAD	0.905	0.789	0.780	0.825	0.739

Respond Letter Table 2: Comparison with other state-of-the-art methods on zero-shot classification task on three downstream tasks. Seen classes denote the diseases that appear in the “entity set” and otherwise unseen classes. AUC scores are reported, and the metrics refer to the macro average on all diseases.

Q 1.3 *The comparison with truly zero-shot models is unfair. As said before, the proposed method is not a truly zero-shot model. First, it is specifically designed for the disease classification task; therefore, it cannot be extended to other tasks, e.g., lesion detection and segmentation. The other zero-shot models compared in Table 1 are general model, not specifically designed for any task; therefore, they may be more versatile. Second, as said above, the authors reported the performance of the proposed model on both seen and unseen diseases, without clearly indicating which diseases are seen and which are unseen. Therefore, the comparison is unfair. The authors just limit the comparison to unseen diseases for a fair comparison with those zero-shot models.*

Reply: In the following, we reply to this question point-by-point, including the evaluation of lesion detection and segmentation, and the evaluation on seen and unseen diseases.

Lesion detection and segmentation. we followed the suggestion of Reviewer 2 to evaluate the model’s localization ability on ChestX-Det10 [5]. Here, we report the grounding performance with **Pointing Game** [8], specifically, pointing game measures how accurate the most confident region in the predicted heatmap is with respect to the ground-truth bounding box. In detail, we extract the region with max response from the output heatmap, and if the region hit the ground-truth mask, it is considered a positive prediction, otherwise negative. Finally, accuracy can be calculated as the pointing game score. As shown in Respond Letter Table 3, KAD surpasses all existing approaches on the average pointing game score.

Evaluation on seen and unseen diseases. We will report performance for seen and unseen diseases separately, as detailed in the previous response.

Method	Atel.	Calc.	Cons.	Effu.	Emph.	Fibr.	Frac.	Mass	Nodu.	Pneu.	Mean
GLoRIA [3]	0.479	0.053	0.737	0.528	0.667	0.366	0.013	0.533	0.156	0.143	0.367
BioViL [1]	0.375	0.105	0.664	0.615	0.795	0.378	0.013	0.500	0.13	0.229	0.380
KAD	0.646	0.132	0.699	0.618	0.644	0.244	0.199	0.267	0.316	0.143	0.391
KAD-512	0.729	0.158	0.713	0.738	0.769	0.293	0.237	0.433	0.377	0.171	0.462
KAD-1024	0.771	0.316	0.692	0.560	0.718	0.379	0.132	0.367	0.571	0.343	0.485

Respond Letter Table 3: Comparison with other state-of-the-art methods on zero-shot grounding task on ChestX-Det10. Pointing game scores are reported. We use the first four letters to represent one disease.

Q 1.4 The authors only evaluated with the ResNet50 backbone as the image encoder, which is a quite old model. Recently, various Transformer models have been proposed and achieved superiority performance. The authors are encouraged to evaluate with more-recent backbones to verify if the findings of this work still hold.

Reply: We perform experiments with ViT-16 [2] as backbone. As shown in Respond Letter Table 4, performance with ResNet and ViT are comparable, showing that the proposed knowledge-enhanced representation learning is robust to the selection of image encoder. We have added the results of KAD with ViT as the backbone for a more comprehensive analysis in the revised manuscript.

Image Encoder	Input size	ChestXray14				CheXpert			
		AUC	MCC	F1	ACC	AUC	MCC	F1	ACC
ResNet-50	224	0.789	0.280	0.323	0.816	0.905	0.589	0.647	0.875
ViT-16	224	0.785	0.276	0.321	0.807	0.907	0.575	0.647	0.884

Respond Letter Table 4: Ablation study of KAD with different image encoders. AUC, MCC, F1, and ACC scores are reported, and the metrics all refer to the macro average on all diseases.

Respond Letter Figure 1: Zero-shot visualization of randomly chosen samples from ChestX-Det10, we present both the original image and attention maps generated from KAD, KAD-512 and KAD-1024. In the original images, red boxes denote lesion areas annotated by radiologists. In the attention maps, the red to blue spectrum are plotted on the original image with red representing high attention regions and blue representing low attention.

Reviewer 2

Q 2.1 *The deep learning model takes resized 224x224 pixel images as inputs and everything were learned from this resolution. The image resolution of 224x224 was adopted because of the original imageNet images? Even using ResNet-50, only the convolutional layers are usable anyway so any image resolution should be able to be handled. The FC or other decision layers need to be re-trained anyway. At 224x224, each pixel is about 2mm to 3mm size (the original X-ray pixel size is about 0.1mm). Patients take 3000x4000 x-ray images for a valid reason and many diseases (especially subtle ones) can be seen clearly if the image size is smaller than 1000 pixels (see the ChestXray14 CVPR 2017 paper). This chest xray screening population are expected to have significant appearance of spatially subtle diseases. Author should investigate using 1024x1024 image resolutions. 224x224 has essential limitations which should be obvious.*

Reply: Thanks for the suggestion. In the original paper, we use the resized 224x224 pixel images as inputs for a fair comparison with existing SOTA methods. Here, we perform experiments with different image resolutions and summarize the results in Respond Letter Table 5. We also present the zero-grounding results on the ChestX-Det10 in Respond Letter Table 3. The ability to localize disease is significantly improved with the image resolution increasing, especially for some diseases with small bounding boxes, KAD-1024 shows not only higher accuracy but also better precision, as shown in Respond Letter Figure 1. We will add the zero-shot results of **KAD-1024** in the revision.

Image Encoder	Input size	CheXpert	ChestXray14		PadChest	
		Seen classes	Seen classes	Unseen classes	Seen classes	Unseen classes
ResNet-50	224	0.905	0.789	0.780	0.825	0.739
ResNet-50	512	0.908	0.791	0.748	0.832	0.762
ResNet-50	1024	0.903	0.804	0.770	0.840	0.757

Respond Letter Table 5: Comparison with other state-of-the-art methods on zero-shot classification task on three downstream tasks. Seen classes denote the diseases that appear in the Entity set and otherwise unseen classes, please refer to the reply of **Q1.2** for more details. AUC scores are reported, and the metrics refer to the macro average on all diseases.

Q 2.2 *Authors should perform more rigorous experiments on localization of the diseases. As now, there is only one figure with a few examples. This is not enough. If the AI cannot localize the disease, it can not be trusted for its diagnosis. <https://arxiv.org/pdf/2010.10298.pdf> <https://arxiv.org/abs/2006.10550> The above dataset can serve the purpose of localization.*

Reply: Refer to Reply for **Q1.3**.

Q 2.3 *The AI system has high or higher false positive rates on classifying healthy patients? The majority population in a clinical setting where chest X-ray exams are performed may be healthy patients.*

Reply: Here, we perform an analysis on KAD’s performance for classifying healthy patients in PadChest, which consists of only 12694 healthy patients out of a total of 39053 patients, *i.e.*,

the dataset we are evaluating is actually biased towards unhealthy patients (67.5% of the total samples). We report the confusion matrix for classification in Respond Letter Table 6. We compute two metrics, namely recall (the ratio between the number of correctly predicted patients and the total number of ‘real’ patients), and false positive rates (the ratio between the number of healthy people wrongly categorized as patients and the total number of healthy people).

$$\text{Recall} = \frac{TP}{FN + TP} = \frac{19860}{3494 + 19860} = 0.850, \quad \text{FPR} = \frac{FP}{FP + TN} = \frac{6499}{6499 + 9200} = 0.414$$

Confusion Matrix		Ground Truth	
		health	disease
KAD	health	9200 True Negative	3494 False Negative
	disease	6499 False Positive	19860 True Positive

Respond Letter Table 6: Confusion Matrix.

As can be seen, KAD achieves high recall on this dataset, which is critical for an AI-based medical auto-diagnosis system to not miss any unhealthy cases.

Reviewer 3

Q 3.1 *Instead of pretraining on raw report text, the proposed method used radiology findings produced by RadGraph (5.4.3 "entity extraction"). This is problematic as RadGraph is a supervised method trained on reports annotated with common findings by board-certified radiologists. So effectively, the proposed method was pretrained with findings extracted by a supervised method, which made the comparison with other truly self-supervised approaches (pretrained on image-report pairs only) not very meaningful. This is the most serious concern about the proposed method, which undermines the central claim that the superiority in the proposed pretraining stems from incorporating domain knowledge. To evaluate truly self-supervised performance, the authors should not use RadGraph or any supervised entity extraction systems at all, but instead start from raw report text.*

Reply: We apologize for the confusing use of “self-supervised” and “pre-training” in the submitted manuscript, which we have revised to manuscript and make clarification.

To start with, we would like to clarify the core of our study, that is, to develop a scalable pre-training procedure for incorporating existing medical domain knowledge, *e.g.*, UMLS, RadGraph, that naturally equips KAD with additional information than existing self-supervised approaches. (1) We agree with the reviewer that training RadGraph [4] would require manual annotation from clinicians, and using it for pre-processing reports would potentially incur extra supervision than existing self-supervised approaches. In the revised manuscript, we treat RadGraph as part of expert knowledge injection, same role as using the UMLS knowledge graph, we will tune the tone of the paper, only use the term ‘knowledge-enhanced pre-training’, rather than ‘self-supervised learning’. (2) We agree that our model benefit from RadGraph while evaluating on seen classes, however, **the evaluation of unseen diseases (not appearing in the “entity set”) is still fair**, as

RadGraph does not provide labels for them. As indicated by the results on PadChest (Respond Letter Table 5), our method largely outperforms previous methods. KAD achieves an AUC of at least 0.900 on 31 classes and at least 0.700 on 111 classes out of 177 **unseen** classes in the PadChest test dataset ($n = 39,053$). (3) We have added a detailed description on the results comparison, see line 88-93 and 172-181. From a practical perspective, KAD training only exploits off-the-shelf tools, thus is equally scalable to large datasets as self-supervised learning approaches do, while demonstrating superior performance on identifying diseases not encountered at training time, and handling long-tail recognition problem. We conduct extensive ablation studies to analyze the contribution of certain model components, and the impact of image resolution and visual backbone.

Overall, we still believe if using the off-the-shelf RadGraph (trained with minimal manual annotation) can bring significant benefits, especially for unseen classes, is a meaningful investigation.

Method	Average AUC	AUC(≥ 0.900)	AUC(≥ 0.700)
ConVIRT [9]	0.515	0	2
GLoRIA [3]	0.486	0	4
BioVIL [1]	0.645	5	59
CheXzero [6]	0.666	12	71
MedKLIP [7]	0.636	5	55
KAD	0.741	31	111

Respond Letter Table 7: Comparison with other state-of-the-art methods on zero-shot classification task on PadChest test set. We report the results on 177 **unseen** classes in the PadChest test dataset ($n = 39,053$).

Q 3.2 *Details about knowledge-enhanced pretraining are also a bit unclear. I have a hard time figuring out what DQN does in pretraining. By definition, there is no annotated disease for an image-report pair. Where would the ground-truth label come from to compute cross entropy loss.*

Reply: As we are using RadGraph, the labels indicating the existence of the entity can thus be obtained from the uncertainty level. However, **note that**, to properly evaluate the zero-shot performance on unseen classes, we only select the top $Q = 40$ most commonly appearing observation entities from the entire reports corpus, denoted as an entity set $\mathcal{Q} = \{q_1, q_2, \dots, q_Q\}$, and get potentially noisy labels with RadGraph. At training time, DQN is optimized to predict the existence of the query entity by the cross entropy loss. And during inference, the query entity is not limited to the entity set, as the relations between medical entities have been implicitly modeled in textural embedding space.

Q 3.3 *Eq 6 is also a bit confusing. The contrastive loss should try to pull corresponding pair closer so I expect x_i and t_i in the numerator. Instead, the equation has x_i/t_j , t_i/x_j . I don't see any explanation about j and relation between i and j .*

Reply: Thank you for pointing it out. We have revised the related symbols as follows.

$$\mathcal{L}_{\text{contrast}} = -\left(\log \frac{e^{(\hat{\mathbf{x}}_i, \hat{\mathbf{t}}_i)/\tau}}{\sum_{k=1}^N e^{(\hat{\mathbf{x}}_i, \hat{\mathbf{t}}_k)/\tau}} + \log \frac{e^{(\hat{\mathbf{t}}_i, \hat{\mathbf{x}}_i)/\tau}}{\sum_{k=1}^N e^{(\hat{\mathbf{t}}_i, \hat{\mathbf{x}}_k)/\tau}}\right). \quad (1)$$

References

- [1] Benedikt Boecking, Naoto Usuyama, Shruthi Bannur, Daniel C Castro, Anton Schwaighofer, Stephanie Hyland, Maria Wetscherek, Tristan Naumann, Aditya Nori, Javier Alvarez-Valle, et al. Making the most of text semantics to improve biomedical vision–language processing. In *European conference on computer vision*, pages 1–21, 2022. Official Implementation: <https://github.com/microsoft/hi-ml/tree/main/hi-ml-multimodal>.
- [2] Alexey Dosovitskiy, Lucas Beyer, Alexander Kolesnikov, Dirk Weissenborn, Xiaohua Zhai, Thomas Unterthiner, Mostafa Dehghani, Matthias Minderer, Georg Heigold, Sylvain Gelly, et al. An image is worth 16x16 words: Transformers for image recognition at scale. *arXiv preprint arXiv:2010.11929*, 2020.
- [3] Shih-Cheng Huang, Liyue Shen, Matthew P Lungren, and Serena Yeung. Gloria: A multimodal global-local representation learning framework for label-efficient medical image recognition. In *Proceedings of the IEEE/CVF International Conference on Computer Vision*, pages 3942–3951, 2021. Official Implementation: <https://github.com/marshuang80/gloria>.
- [4] Saahil Jain, Ashwin Agrawal, Adriel Saporta, Steven Truong, Du Nguyen Duong, Tan Bui, Pierre Chambon, Yuhao Zhang, Matthew Lungren, Andrew Ng, Curtis Langlotz, Pranav Rajpurkar, and Pranav Rajpurkar. Radgraph: Extracting clinical entities and relations from radiology reports. In J. Vanschoren and S. Yeung, editors, *Proceedings of the Neural Information Processing Systems Track on Datasets and Benchmarks*, volume 1, 2021.
- [5] Jingyu Liu, Jie Lian, and Yizhou Yu. Chestx-det10: Chest x-ray dataset on detection of thoracic abnormalities, 2020.
- [6] Ekin Tiu, Ellie Talius, Pujan Patel, Curtis P Langlotz, Andrew Y Ng, and Pranav Rajpurkar. Expert-level detection of pathologies from unannotated chest x-ray images via self-supervised learning. *Nature Biomedical Engineering*, pages 1–8, 2022.
- [7] Chaoyi Wu, Xiaoman Zhang, Ya Zhang, Yanfeng Wang, and Weidi Xie. Medklip: Medical knowledge enhanced language-image pre-training. *medRxiv*, 2023.
- [8] Jianming Zhang, Sarah Adel Bargal, Zhe Lin, Jonathan Brandt, Xiaohui Shen, and Stan Sclaroff. Top-down neural attention by excitation backprop. *International Journal of Computer Vision*, 126(10):1084–1102, 2018.
- [9] Yuhao Zhang, Hang Jiang, Yasuhide Miura, Christopher D Manning, and Curtis P Langlotz. Contrastive learning of medical visual representations from paired images and text. In *Machine Learning for Healthcare*, 2022. Highest Starred Implementation: <https://github.com/edreisMD/ConVIRT-pytorch>.

REVIEWER COMMENTS

Reviewer #1 (Remarks to the Author):

The revision has addressed all my concerns and I am happy to recommend accepting this manuscript as it is.

Reviewer #2 (Remarks to the Author):

I think authors overall did a good job on answering our questions and addressed the questions well in the revised manuscript. I would support publishing this article at Nature Communications. I have no further major questions.

Reviewer #3 (Remarks to the Author):

Thanks authors for the revision and detailed response!

A key concern for the prior submission is that it incorrectly claims to propose a self-supervised method while in effect using the RadGraph tool which was trained using a significant amount of supervised labels. The authors acknowledged this in the response, which is a step in the right direction:

"We apologize for the confusing use of "self-supervised" and "pre-training" in the submitted manuscript, which we have revised to manuscript and make clarification. To start with, we would like to clarify the core of our study, that is, to develop a scalable pretraining procedure for incorporating existing medical domain knowledge, e.g., UMLS, RadGraph, that naturally equips KAD with additional information than existing self-supervised approaches."

However, the revision didn't quite address this problem. The empirical comparison with other self-supervised learning methods still leaves several key questions open.

First, the authors still claim that the proposed method uses "minimal manual annotation". E.g., in both Intro and Discussion, the paper states that: "In this paper, we aim to build a foundation model for chest X-rays with minimal manual annotation."

This is incorrect and raises the same concern as above. RadGraph is a fully supervised model that requires significant manual annotation effort, and the annotation is directly related to the evaluation tasks (e.g., both papers evaluated on Chexpert). By using RadGraph as a core resource, the pretraining procedure can not be considered as being pretrained with "minimal manual annotation". The authors should make it very clear upfront that the proposed method relied on RadGraph and there is significant manual annotation effort required to train it.

The authors suggested that the use of RadGraph is similar to that of UMLS. While it is true to both require manual efforts to construct, they are not the same. UMLS is an ontology (or a collection of ontologies with cross-linking). By contrast, RadGraph is not a static knowledge graph, but a supervised annotation tool that can identify radiology-specific entities and relations from clinical text (by resolving ambiguity and variation). UMLS has been used in clever ways to self-supervise entity extraction (e.g., KRISBERT, EMNLP 2022), but it doesn't have annotation capabilities like RadGraph out of box.

The characterization of RadGraph as a named entity recognition (NER) tool is also a bit misleading. A key contribution in RadGraph is to not only identify entities, but also assert their presence and anatomic relations. E.g., note the difference between a mention of effusion (may be present, may be not) vs an assertion of the presence of effusion.

So, the paper should not lump UMLS with RadGraph and claim that the proposed method is a generic method for leveraging (static) knowledge graph. The two are only related insofar as they are both preexisting resources.

As the authors have clarified, RadGraph is used in multiple places (e.g., entity extraction; DQN). So the ablation studies should be done to highlight the effect of incorporating RadGraph (vs not). E.g., an ideal ablation should start with the proposed method w/o using any pre-existing resources (i.e., fully comparable to self-supervised learning), and then go on to add UMLS, and UMLS + RadGraph.

Another required baseline would be to take a frozen self-supervised model and augment it with RadGraph. I.e., the difference from the proposed method is that RadGraph is not used during pretraining to shape the embedding, but it's still available at inference time. This would be a good baseline to truly assess how much one gains from incorporating RadGraph annotation in pretraining.

The authors claim that there are many unseen classes in MIMIC-CXR. E.g.:

"

CheXpert: collected from Stanford Hospital, all 5 classes are seen at training time;

ChestXray14: collected from NIH Clinical Center, with 15 seen classes (14 seen and only 1 unseen, fibrosis);

PadChest: collected from Indiana University, with a total of 193 classes (16 seen and 177 unseen).

"

If such unseen entities are truly absent from RadGraph annotation, then the evaluation on such classes is indeed more compelling. Given the large scale of MIMIC-CXR for chest X-ray, I found it a bit surprising that MIMIC-CXR would miss so many classes (e.g., 177 out of 193 in PadChest). The authors might want to clarify exactly how such classes are determined. What are examples of such missing classes? Did RadGraph miss them completely, or still cover a related (e.g., more abstract) concept?

Respond to Review: NCOMMS-23-03142-T

Knowledge-enhanced Visual-Language Pre-training on Chest Radiology Images

Anonymous submission

We thank all reviewers for their valuable comments. In the following, we aim to resolve their concerns and will update the final manuscript accordingly. All codes and models will be released.

Reviewer 1

Q 1.1 The revision has addressed all my concerns and I am happy to recommend accepting this manuscript as it is.

Reply: Thank you very much for taking the time to review our manuscript and for your valuable feedback. We are delighted to learn that the revision has successfully addressed all your concerns. We sincerely appreciate your efforts in providing us with insightful comments, which have greatly contributed to the improvement of our work.

Reviewer 2

Q 2.1 I think authors overall did a good job on answering our questions and addressed the questions well in the revised manuscript. I would support publishing this article at Nature Communications. I have no further major questions.

Reply: We greatly appreciate your thorough review of our manuscript and your positive feedback. Thank you for acknowledging our efforts in addressing your questions and concerns in the revised version. Your valuable input and guidance throughout the review process have been immensely valuable to us, and we are sincerely thankful for your support.

Reviewer 3

Q 3.1 Misleading claim of “minimal manual annotation”. First, the authors still claim that the proposed method uses “minimal manual annotation”. E.g., in both Intro and Discussion, the paper states that: “In this paper, we aim to build a foundation model for chest X-rays with minimal manual annotation.” This is incorrect and raises the same concern as above. RadGraph is a fully supervised model that requires significant manual annotation effort, and the annotation is directly related to the evaluation tasks (e.g., both papers evaluated on Chexpert). By using RadGraph as a core resource, the pretraining procedure can not be considered as being pretrained with “minimal manual annotation”. The authors should make it very clear upfront that the proposed method relied on RadGraph and there is significant manual annotation effort required to train it.

Reply: We apologize for any confusion caused by the statement in our introduction and discussion sections that our method uses “minimal manual annotation”. We have made revisions to the manuscript to clarify that RadGraph is a fully supervised model that requires significant manual annotation effort. However, we would like to highlight that while RadGraph serves as a valuable tool for identifying radiology-specific entities and asserting their presence and anatomical relations from clinical text, we conduct experiments to show that RadGraph is not **indispensable**.

In KAD, we specifically utilize the extracted entities and their presence, which can also be accomplished using alternative resources such as UMLS or the latest language model, ChatGPT [1], as exemplified in Reply for **Q3.3**.

Q 3.2 Mischaracterization of RadGraph. *The authors suggested that the use of RadGraph is similar to that of UMLS. While it is true to both require manual efforts to construct, they are not the same. UMLS is an ontology (or a collection of ontologies with cross-linking). By contrast, RadGraph is not a static knowledge graph, but a supervised annotation tool that can identify radiology-specific entities and relations from clinical text (by resolving ambiguity and variation). UMLS has been used in clever ways to self-supervise entity extraction (e.g., KRISBERT, EMNLP 2022), but it doesn't have annotation capabilities like RadGraph out of box. The characterization of RadGraph as a named entity recognition (NER) tool is also a bit misleading. A key contribution in RadGraph is to not only identify entities, but also assert their presence and anatomic relations. E.g., note the difference between a mention of effusion (may be present, may be not) vs an assertion of the presence of effusion. So, the paper should not lump UMLS with RadGraph and claim that the proposed method is a generic method for leveraging (static) knowledge graph. The two are only related insofar as they are both preexisting resources.*

Reply: We agree that RadGraph and UMLS are not the same, and we apologize for any confusion that our statement of these two resources may have caused misleading. We have revised our manuscript to reflect this distinction and avoid any potential confusion. First, we revise the description of RadGraph in Section 4 (lines 414-417). The revision is shown as follows.

“We utilized the off-the-shelf RadGraph, for identifying radiology-specific entities and asserting their presence and anatomical relations from clinical text. RadGraph was trained on 500 radiology reports from the MIMIC-CXR dataset annotated by board-certified radiologists. ”

Second, we have modified the “off-the-shelf named entity recognition (NER) toolbox” to “off-the-shelf reports information extraction toolbox” in our paper, and clarify that RadGraph is a tool capable of automatically generating knowledge graphs from radiology reports based on the structural prior in reports.

Q 3.3 Ablation study of RadGraph. *As the authors have clarified, RadGraph is used in multiple places (e.g., entity extraction; DQN). So the ablation studies should be done to highlight the effect of incorporating RadGraph (vs not). E.g., an ideal ablation should start with the proposed method w/o using any pre-existing resources (i.e., fully comparable to self-supervised learning), and then go on to add UMLS, and UMLS + RadGraph. Another required baseline would be to take a frozen self-supervised model and augment it with RadGraph. I.e., the difference from the proposed method is that RadGraph is not used during pretraining to shape the embedding, but it's still available at inference time. This would be a good baseline to truly assess how much one gains from incorporating RadGraph annotation in pretraining.*

Reply: Thanks for the suggestion. To analyze the effect of incorporating RadGraph (vs not), we conduct an ablation experiment on KAD without the use of RadGraph. The main difference is in the entity extraction section, as the supervision of DQN is also derived from this module.

Respond Letter Figure 1: Overview of entity extraction with UMLS.

Entity Extraction with UMLS. We utilized the Unified Medical Language System (UMLS) for entity extraction as illustrated in Fig. 1. For each sentence in the radiology report, we were able to extract a sequence of entities (entity, concept, CUI, TUI) with `spacy`, where CUI represents “Concept Unique Identifier” and TUI represents “Type Unique Identifier”. Here, we provide the pseudo-code as follows:

```
import spacy
from collections import defaultdict
# "en_core_sci_lg" is a full spacy pipeline for biomedical data with a larger
# vocabulary and 600k word vectors
nlp = spacy.load("en_core_sci_lg")
nlp.add_pipe("abbreviation_detector")
# "scispacy_linker" is a spacy component that performs linking to a knowledge
# base~(UMLS in our case)
nlp.add_pipe("scispacy_linker",
             config={"resolve_abbreviations": True, "linker_name": "umls"})
sentence_entities = [] #sentence_entities refers the extracted list
caption_nlp = nlp(caption) # caption refers to the sentence of the radiology
# report
entities = caption_nlp.ents
for entity in entities:
    entity_dict = defaultdict(list)
    linker = nlp.get_pipe("scispacy_linker")
    for umls_ent in entity._.kb_ents:
        umls_ent_info = linker.kb.cui_to_entity[umls_ent[0]]
        entity_dict['entity'] = entity
        entity_dict['concept'] = umls_ent_info[1]
        entity_dict['CUI'] = umls_ent_info[0]
        entity_dict['TUI'] = umls_ent_info[3]
    sentence_entities.append(entity_dict)
```

We filtered the entity list for each sentence based on TUI, retaining only the entities with TUI in (T033: Finding, and T047: Disease or Syndrome). Next, we match these entities with our entity set $\mathcal{Q} = \{q_1, q_2, \dots, q_Q\}$, except for ‘normal’. To determine the presence of an entity, we adopted a straightforward rule: if the sentence contains the words ‘no’, ‘none’, or ‘normal’, the label is set to 0 (absent); otherwise, the label is set to 1 (present). For entities that are not mentioned in the report, we assigned a label of 0. If all entities have a label of 0, we set the label of ‘normal’ to

1. To this end, we replaced all the parts where RadGraph was used in our paper with the use of UMLS, and the result is shown in Table 1 (KAD w/o RadGraph).

Respond Letter Figure 2: Overview of entity extraction with UMLS.

Entity Extraction with ChatGPT. Large language models (LLMs), such as ChatGPT [1] have recently demonstrated remarkable progress in a wide range of natural language processing (NLP) tasks. Here, we also conduct an ablation experiment by using ChatGPT for entity extraction. As shown in Fig. 2, we input the radiology report as the content to the ChatGPT and employ the following prompt.

For the given report of a chest x-ray, determine if the patient has any of the following findings: finding_list = [pleural effusion, opacity, pneumothorax, edema, atelectasis, tube, consolidation, enlarged cardiomeastinum, tip, pneumonia, line, cardiomegaly, fracture, calcification, medical device, engorgement, nodule, wire, pacemaker, pleural thicken, marking, scar, hyperinflate, blunt, collapse, emphysema, aerate, mass, infiltration, obscure, deformity, hernia, drainage, distention, shift, lesion, hardware, dilation, aspiration]. The output should use the following template: label_list = [i if finding exists for finding in finding_list]. If no finding exists, output label_list = []

We then process the output label_list to one-hot label, and the result is shown in Table 1 (KAD w/ ChatGPT).

Image Encoder	ChestXray14				CheXpert			
	AUC	MCC	F1	ACC	AUC	MCC	F1	ACC
Ablation on proposed modules								
KAD w/o Stage1	0.752	0.228	0.274	0.748	0.894	0.546	0.620	0.858
KAD w/o random select	0.751	0.242	0.290	0.780	0.878	0.571	0.671	0.812
KAD w/o DQN	0.672	0.144	0.109	0.747	0.822	0.419	0.508	0.806
Ablation on entity extraction module								
w/ UMLS	0.773	0.268	0.308	0.833	0.904	0.562	0.635	0.867
w/ ChatGPT	0.784	0.284	0.336	0.845	0.887	0.573	0.622	0.888
w/ RadGraph	0.789	0.280	0.323	0.816	0.905	0.589	0.647	0.875

Respond Letter Table 1: Ablation study of KAD with different image encoders. AUC, MCC, F1, and ACC scores are reported, and the metrics all refer to the macro average on all diseases.

As indicated by the results, RadGraph is indeed a valuable tool that has demonstrated its effectiveness in our research. We acknowledge its significant contribution to our work and recognize its potential for further improvement. It is also crucial to highlight that the performance of our

model is not solely reliant on RadGraph, as shown in the ablation study, both KAD w/o RadGraph and KAD w/ ChatGPT work similarly well. That shows the critical component in KAD is our proposed DQN architecture, while different ways for extracting entities work similarly. We will add the ablation experiments of entity extraction in the revision.

Q 3.4 Explanation on unseen classes. *The authors claim that there are many unseen classes in MIMIC-CXR. If such unseen entities are truly absent from RadGraph annotation, then the evaluation on such classes is indeed more compelling. Given the large scale of MIMIC-CXR for chest X-ray, I found it a bit surprising that MIMIC-CXR would miss so many classes (e.g., 177 out of 193 in PadChest). The authors might want to clarify exactly how such classes are determined. What are examples of such missing classes? Did RadGraph miss them completely, or still cover a related (e.g., more abstract) concept?*

Reply: To clarify, at training stage, we selected the top Q most commonly appearing observation entities from the entire reports corpus as the “entity set”. The diseases that appeared in the “entity set” (obtained from MIMIC-CXR) were considered as seen diseases, while the others were considered unseen diseases. As we presented in our manuscript, specifically in lines 397-401, “After extracting all entities from the reports, we select the top Q most commonly appearing observation entities from the entire reports corpus, denoted as an entity set $\mathcal{Q} = \{q_1, q_2, \dots, q_Q\}$, and get a label indicating the existence of the entity from the uncertainty level.” The model is only trained for these Q entities from the “entity set”. Therefore, when we refer to unseen classes, it does not imply that RadGraph completely missed these classes. Rather, it means that these classes have not been learned by the DQN because they are not included in the entity set. We revise the definition of unseen classes in line to make clarify, the revision is shown as follows,

“Here, we denote the diseases that are seen by Disease Query Network (DQN) at the model training stage as seen diseases, and the others as unseen diseases.”

In addition, the class definitions in the entity set are more abstract, while the Padchest dataset contains more fine-grained and long-tailed class distributions. For example, in the “entity set”, we have a category for ‘tube’. However, in Padchest, there are specific types of tubes, such as endotracheal tubes, NSG tubes, tracheostomy tubes, chest drain tubes, ventriculoperitoneal drain tubes, nephrostomy tubes, and gastrostomy tubes. These specific tube types are not included in the entity set and are considered unseen by RadGraph. In many fine-grained recognition tasks [3, 2], similar cases where specific subclasses are not explicitly defined are treated as unseen classes in the evaluation.

However, even if we exclude classes that are related to the “entity set”, such as those listed in the example above, there remain a total of 122 unseen classes. Among these, 16 classes achieve an AUC above 0.9, and 72 classes achieve an AUC above 0.7. Here we provide some example missing classes: [‘artificial mitral heart valve’, ‘sternotomy’, ‘artificial aortic heart valve’, ‘artificial heart valve’, ‘abscess’, ‘pulmonary venous hypertension’, ‘central venous catheter via subclavian vein’, ‘central venous catheter via jugular vein’, ‘round atelectasis’, ‘heart insufficiency’, ‘lymphangitis carcinomatosa’, ‘central venous catheter via umbilical vein’].

We hope that this explanation addresses your question regarding the presence of unseen classes in our evaluation.

References

[1] Openai. introducing chatgpt. <https://openai.com/blog/chatgpt/>, 2023.

- [2] Xiuye Gu, Tsung-Yi Lin, Weicheng Kuo, and Yin Cui. Open-vocabulary object detection via vision and language knowledge distillation. In *International Conference on Learning Representations*, 2022.
- [3] Dat Huynh and Ehsan Elhamifar. Compositional zero-shot learning via fine-grained dense feature composition. *Advances in Neural Information Processing Systems*, 33:19849–19860, 2020.

REVIEWERS' COMMENTS

Reviewer #3 (Remarks to the Author):

Thanks authors for their detailed responses and improvement to the paper. This revision has addressed all my concerns.